# QCircuitNet: A Large-Scale Hierarchical Dataset for Quantum Algorithm Design

## Abstract

Quantum computing is an emerging field recognized for the significant speedup it offers over classical computing through quantum algorithms. However, designing and implementing quantum algorithms pose challenges due to the complex nature of quantum mechanics and the necessity for precise control over quantum states. Despite the significant advancements in AI, there has been a lack of datasets specifically tailored for this purpose. In this work, we introduce QCircuitNet, the first benchmark and test dataset designed to evaluate AI's capability in designing and implementing quantum algorithms in the form of quantum circuit codes. Unlike using AI for writing traditional codes, this task is fundamentally different and significantly more complicated due to highly flexible design space and intricate manipulation of qubits. Our key contributions include:

1. A general framework which formulates the key features of quantum algorithm design task for Large Language Models.

2. Implementation for a wide range of quantum algorithms from basic primitives to advanced applications, with easy extension to more quantum algorithms.

3. Automatic validation and verification functions, allowing for iterative evaluation and interactive reasoning without human inspection.

4. Promising potential as a training dataset through primitive fine-tuning results.

We observed several interesting experimental phenomena: fine-tuning does not always outperform few-shot learning, and LLMs tend to exhibit consistent error patterns. QCircuitNet provides a comprehensive benchmark for AI-driven quantum algorithm design, offering advantages in model evaluation and improvement, while also revealing some limitations of LLMs in this domain.

## 1 Introduction

Quantum computing is an emerging field in recent decades because algorithms on quantum computers may solve problems significantly faster than their classical counterparts. From the perspective of theoretical computer science, the design of quantum algorithms have been investigated in various research directions - see the survey (Dalzell et al., 2023) and the quantum algorithm zoo (Zoo, 2024). However, the design of quantum algorithms on quantum computers has been completed manually by researchers. This process is notably challenging due to highly flexible design space and extreme demands for a comprehensive understanding of mathematical tools and quantum properties.

For these reasons, quantum computing is often considered to have high professional barriers. As the discipline evolves, we aim to explore more possibilities for algorithm design and implementation in the quantum setting. This is aligned with recent advances among AI for Science, including AlphaFold (Jumper et al., 2021), AlphaGeometry (Trinh et al., 2024), etc. Recently, large language models (LLMs) have also become widely applicable among AI for science approaches (Yang et al., 2024; Zhang et al., 2024; Yu et al., 2024). LLMs represent the best practice of sequential modeling methods at current stage. They have an edge over other models in possessing abundant pre-training knowledge and providing human-friendly interfaces which support human-machine collaboration. Therefore, we gear LLMs for quantum algorithm design.

As far as we know, there has not been any dataset for AI in quantum algorithm design. Existing work combining quantum computing and AI mostly targets at exploiting quantum computing for

AI; there are some papers applying AI for quantum computing, but they either consider niche problems (Nakayama et al., 2023; Schatzki et al., 2021) or limited functions (Tang et al., 2023; Fürrutter et al., 2024), not quantum algorithm datasets of general interest (see Section 2). However, unlike classical code generation where abundant data exist, the most challenging aspect for quantum algorithm design is the lack of sufficient data, and hence the difficulty of generalization in training AI models. Therefore, datasets for quantum algorithm design are solicited.

Descriptions of quantum algorithms in natural language could be verbose and vague. Mathematical formulas, while precise and succinct, are difficult to verify automatically. To accommodate with LLMs, we make a change of perspective by formulating quantum algorithms as programming languages. This allows for precise representation of a quantum algorithm, enables automatic verification procedure, and bridges the gap between theoretical design and circuit implementations. Furthermore, meaningful quantum algorithms which can be efficiently implemented have no more than polynomially many gates (Poulin et al., 2011), and thus such formulations have the theoretical benefits allowing for scalable representations.

**Key Contributions.** In this work, We propose QCircuitNet, the first comprehensive, structured dataset for quantum algorithm design. Technically, QCircuitNet has the following key contributions:

- It formulates the task for Large Language Models (LLMs) with a carefully designed framework encompassing the key features of quantum algorithm design, including problem description, quantum circuit codes, classical post-processing, and verification functions. It maintains the black-box nature of oracles and characterizes query complexity properly.

- It implements a wide range of quantum algorithms from basic primitives and textbook-level algorithms to advanced applications. We demonstrate the compatibility with complex algorithms through Generalized Simon's Problem and showcase the easy extension to advanced algorithms.

- It has automatic validation and verification functions, allowing for iterative evaluation without human inspection. This further enables interactive reasoning which may improve the performance.

- It showcases the potential as a training dataset through primitive fine-tuning results. As we expand the dataset to include more algorithms and explore novel fine-tuning methods, it will hopefully contribute to interactive quantum algorithm design and implementation significantly.

## 2 RELATED WORK

**Quantum Machine Learning.** To the best of our knowledge, QCircuitNet is the first dataset tailored specifically for quantum algorithm design. Previous efforts combining quantum computing with AI primarily fall under the category of Quantum Machine Learning (QML), which aims at leveraging the unique properties of quantum systems to enhance machine learning algorithms and achieve improvements over their classical counterparts (Schuld et al., 2015; Biamonte et al., 2017; Ciliberto et al., 2018). Corresponding datasets often focus on encoding classical data into quantum states. For instance, MNISQ (Placidi et al., 2023) is a dataset of quantum circuits representing the original MNIST dataset (LeCun et al., 1998) generated by the AQCE algorithm (Shirakawa et al., 2021). Considering the intrinsic nature of quantum properties, another category of datasets focuses on collecting quantum data to demonstrate quantum advantages since classical machine learning methods can fail to characterize particular patterns of quantum data. For example, Nakayama et al. (2023) created a VQE-generated quantum circuit dataset for classification of variational ansatzes and showed its quantum supremacy. NTangled (Schatzki et al., 2021) further investigated different types of entanglement and composed quantum states with various multipartite entanglement for classification. While these datasets successfully demonstrate the supremacy of quantum computing, they address relatively niche problems whose practical applications are unclear.

**AI for Quantum Computing.** This research direction explores the possibility of leveraging AI to facilitate the advancement of quantum computing. QDataSet (Perrier et al., 2022) collects data from simulations of one- and two-qubit systems and targets training classical machine learning algorithms for quantum control, quantum tomography, and noise mitigation. LLM4QPE (Tang et al., 2023) is a large language model style paradigm for predicting quantum system properties with pre-training and fine-tuning workflows. While the paradigm is interesting, the empirical experiments are limited

to two downstream tasks: quantum phase classification and correlation prediction. Fürrutter et al. (2024) studied the application of diffusion models (Sohl-Dickstein et al., 2015; Rombach et al., 2022) to quantum circuit synthesis (Saeedi & Markov, 2013; J. et al., 2022). Scalability issues must be addressed to achieve practical and meaningful unitary compilation through this methodology.

**Quantum Circuit Benchmarks.** The aforementioned works represent meaningful explorations at the intersection of AI and quantum computing. However, none of them considers the task which interests the quantum computing community (from the theoretical side) the most: quantum algorithm design. Our work aims to take the first step in bridging this gap. It is worth noting that several quantum algorithm circuit benchmarks already exist, such as QASMBench (Li et al., 2023), MQTBench (Quetschlich et al., 2023), and VeriQBench (Chen et al., 2022). However, these benchmarks are designed to evaluate the performance of NISQ (Noisy Intermediate-Scale Quantum) (Preskill, 2018) machines or quantum software tools, rather than for training and evaluating AI models. For instance, QASMBench includes a diverse variety of quantum circuits based on OpenQASM representation (Cross et al., 2022), covering quantum circuits with qubit sizes ranging from 2 to 127. However, it fails as a dataset for AI in that it does not capture the design patterns of each algorithm and ignores the post-processing procedure and construction of different oracles, which are crucial to quantum algorithm design. Similar limitations apply to MQTBench and VeriQBench.

## 3 PRELIMINARIES FOR QUANTUM COMPUTING

In this section, we will introduce necessary backgrounds for quantum computing related to this paper. Additional preliminaries can also be found in Appendix B. A more detailed introduction to quantum computing can be found in the standard textbook by Nielsen & Chuang (2000).

**Quantum States.** In classical computing, the basic unit is a bit. In quantum computing, the basic unit is a *qubit*. Mathematically, $n$ ($n \in \mathbb{N}$) qubits forms an $N$-dimensional Hilbert space for $N = 2^n$. An $n$-qubit *quantum state* $|\phi\rangle$ can be written as

$$|\phi\rangle = \sum_{i=0}^{N-1} \alpha_i |i\rangle, \quad \text{where} \quad \sum_{i=0}^{N-1} |\alpha_i|^2 = 1. \tag{1}$$

Here $|\cdot\rangle$ represents a column vector, also known as a ket state. The tensor product of two quantum states $|\phi_1\rangle = \sum_{i=0}^{N-1} \alpha_i |i\rangle$ and $|\phi_2\rangle = \sum_{j=0}^{M-1} \beta_j |j\rangle$ with $M = 2^m$, $m \in \mathbb{N}$ is defined as

$$|\phi_1\rangle \otimes |\phi_2\rangle = \sum_{i=0}^{N-1} \sum_{j=0}^{M-1} \alpha_i \beta_j |i, j\rangle, \tag{2}$$

where $|i, j\rangle$ is an $(n + m)$-qubit state with first $n$ qubits being the state $|i\rangle$ and the last $m$ qubits being the state $|j\rangle$. When there is no ambiguity, $|\phi_1\rangle \otimes |\phi_2\rangle$ can be abbreviated as $|\phi_1\rangle|\phi_2\rangle$.

**Quantum Oracles.** To study a Boolean function $f\colon \{0,1\}^n \to \{0,1\}^m$, we need to gain its access. Classically, a standard setting is to being able to *query* the function, in the sense that if we input an $x \in \{0,1\}^n$, we will get the output $f(x) \in \{0,1\}^m$. In quantum computing, the counterpart is a quantum query, which is instantiated by a *quantum oracle*. Specifically, the function $f$ is encoded as an oracle $U_f$ such that for any $x \in \{0,1\}^n$, $z \in \{0,1\}^m$,

$$U_f|x\rangle|z\rangle = |x\rangle|z \oplus f(x)\rangle, \tag{3}$$

where $\oplus$ is the plus modulo 2. Note that a quantum query to the oracle is stronger than a classical query in the sense that the quantum query can be applied to a state in *superposition*: For an input state $\sum_i c_i |x_i\rangle|z_i\rangle$ with $\sum_i |c_i|^2 = 1$, the output state is $\sum_i c_i |x_i\rangle|z_i \oplus f(x_i)\rangle$; measuring this state gives $x_i$ and $z_i \oplus f(x_i)$ with probability $|c_i|^2$. A classical query for $x$ can be regarded as the special setting with $c_1 = 1$, $x_1 = x$, $z_1 = 0^m$, and $c_i = 0$ for all other $i$.

**Quantum Gates.** Similar to classical computing that can stem from logic synthesis with AND, OR, and NOT, quantum computing is also composed of basic quantum gates. For instance, the Hadamard $H$ is the matrix $\frac{1}{\sqrt{2}} \begin{bmatrix} 1 & 1 \\ 1 & -1 \end{bmatrix}$, satisfying $H|0\rangle = \frac{1}{\sqrt{2}}(|0\rangle + |1\rangle)$ and $H|1\rangle = \frac{1}{\sqrt{2}}(|0\rangle - |1\rangle)$. In general, an $n$-qubit quantum gate is a unitary matrix from $\mathbb{C}^{2^n \times 2^n}$.

# 4 QCircuitNet Dataset

## 4.1 Task Suite

For the general purpose of quantum algorithm design, we consider three categories of tasks: oracle construction, algorithm design, and random circuit synthesis. These three tasks are crucial for devising and implementing quantum algorithms, with oracle construction serving as the premise for algorithm design, and random circuits serving as a main demonstration for quantum supremacy.

### 4.1.1 Task I: Oracle Construction

The construction of such an oracle $U_f$ using quantum gates is deeply rooted in the research topic of reversible quantum logic synthesis, which remains a challenge for complex Boolean functions. In this dataset, we mainly focus on the construction of textbook-level oracles: Bernstein-Vazirani Problem (Bernstein & Vazirani, 1993), Deutsch-Jozsa Problem (Deutsch & Jozsa, 1992), Simon's Problem (Simon, 1997), and Grover's algorithm for unstructured search (Grover, 1996) (including constructions of both the oracle and the diffusion operator). There is another category of more flexible oracle construction tasks which we refer to as "Problem Encoding". For example, one can apply Grover's oracle to solving constraint problems such as SAT and triangle finding (Ambainis, 2004). Formulating problem encoding tasks for LLMs slightly differs from quantum logic synthesis, and we refer the readers to Appendix A.2 for more detailed discussion.

### 4.1.2 Task II: Quantum Algorithm Design

There are several challenges to address in order to formulate quantum algorithm design task precisely:

- From a theoretical perspective, the oracle is usually provided as a blackbox gate since the goal of many algorithms is to determine the property of the function $f(x)$ encoded by the oracle $U_f$. If the model has access to the gate implementation of the oracle, it can directly deduce the property from the circuit, failing the purpose of designing a quantum algorithm to decode the information. However, for all experimental platforms, a quantum circuit needs to be explicitly constructed to compile and run successfully, i.e., the oracle should be provided with exact gate implementation. Most tutorials and benchmarks (especially those based on Qiskit and OpenQASM) simply merge the circuit implementation of the oracle and the algorithm as a whole for demonstration purposes. When we gear LLMs for quantum algorithm design, how to separate the algorithm circuits from oracle implementation to avoid information leakage is a critical point to consider.

- A quantum algorithm constitutes not only the quantum circuit, but also its interpretation of measurement results. For example, in Simon's algorithm, the measurement results $y_i$ are not the direct answer $s$ to the problem, but rather satisfies the property $s \cdot y_i = 0$. Linear equations need to be solved to obtain the final answer. In this case, for a complete algorithm design, the model should also specify the the post-process steps to derive the answer to the original problem.

- Quantum circuits for the same algorithm vary with different qubit number $n$. Although this is trivial for theoretical design, it needs to be considered when implementing concrete quantum circuits.

In this category, we cover a wide range of quantum algorithms with varying complexity, from fundamental primitives and textbook-level algorithms to *advanced applications*. For example, we implemented Generalized Simon's Problem (Ye et al., 2021), a more advanced version of the standard Simon's problem and an active area of research in recent years (Ye et al., 2021; Wu et al., 2022). The setting is formally stated as follows: given an (unknown) function $f: \mathbb{Z}_p^n \to X$ where $X$ is a finite set and a $k$ is a positive integer satisfying $k < n$, it is guaranteed that there exists a subgroup $S \leq \mathbb{Z}_p^n$ of rank $k$ such that for any $x, y \in \mathbb{Z}_p^n, f(x) = f(y)$ iff $x - y \in S$. The goal is to find $S$. Intuitively, the generalized Simon's problem extends the standard Simon's problem from binary to $p$-ary bases and from a single secret string to a subgroup of rank $k$. Beyond universal quantum algorithms, we also consider quantum teleportation and quantum key distribution, two widespread protocols in quantum information. We cover their details in Appendix B.

### 4.1.3 Task III: Random Circuit Synthesis

The third task we consider is random circuit synthesis. On the one hand, random circuit sampling is the first algorithm for showing quantum supremacy by Google in 2019 (Arute et al., 2019), and is still widely applied to demonstrate the power of quantum algorithms in recent research (Wu et al., 2021; Bluvstein et al., 2024; DeCross et al., 2024). On the other hand, this also naturally enlarges our dataset. To be more specific, we generate the dataset by randomly sampling gates from a Clifford gate set {H, S, CNOT} and a universal set {H, S, T, CNOT} separately. We then simulate the circuits to obtain the final quantum states. In each task, the problem description provides the vector of the final state, and the model is required to generate quantum circuits that reproduce this state using the specified gate set.

### 4.2 Dataset Structure

The overall structure of QCircuitNet is illustrated as follows (more details are given in Appendix A):

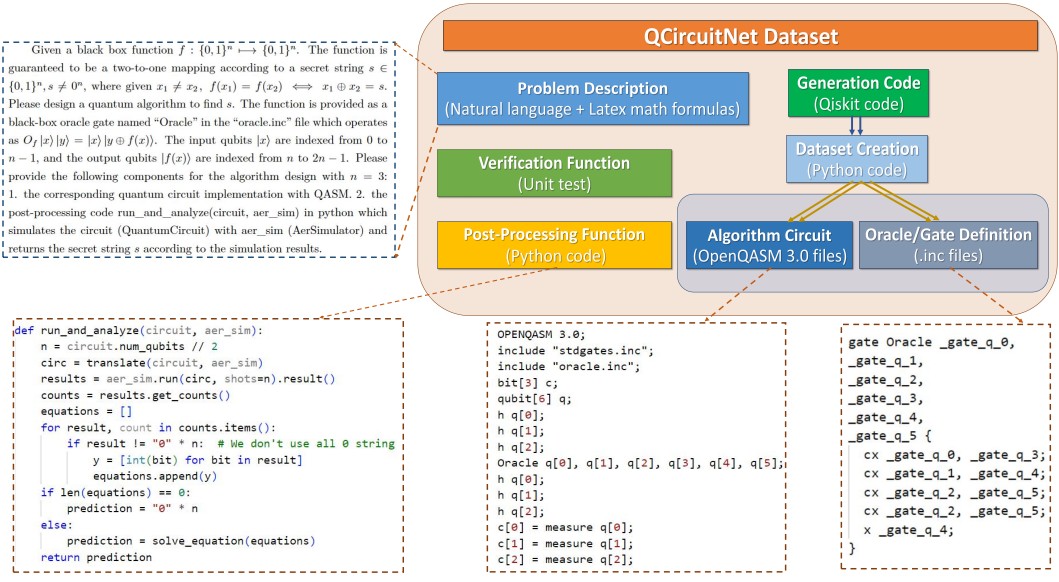

Figure 1: Structure of QCircuitNet. The components of QCircuitNet are presented in the frame on the top-right. As a showcase, this figure presents the components for Simon's problem (Simon, 1997), including its problem description in natural language, post-processing function in python code, circuit in a .qasm file, and oracle definition in a .inc file.

**Design Principles.** As discussed in Section 4.1, a critical consideration in formulating the framework is the dilemma between providing the oracle as a black box for quantum algorithm design and the need for its explicit construction to execute the circuit and interpret the results, making the algorithm design complete. Additionally, model training and reference present challenges, particularly for LLMs in generating complex and precise composite gates and evaluating the results efficiently. To address these obstacles, we highlight the following construction principles, which are specially designed to adapt to these tasks:

- For algorithm design tasks, as discussed in Section 4.1.2, we provide the oracle as a black-box gate named "Oracle" with the explicit definition in a separate "oracle.inc" library, which is supported by the OpenQASM 3.0 grammar. In this way, we make sure that the model can use the oracle without accessing its underlying function, which solves the problem of isolating oracle definition from the algorithm circuit.

- For oracle construction tasks, we ask the model to directly output the quantum circuit in QASM format. For algorithm design task, we require both a quantum circuit and a post-processing function to derive the final answer from circuit execution results. Moreover, we ask the model to explicitly

set the shots needed to run the circuit itself in order to characterize the query complexity, which is critical in the theoretical analysis of algorithms.

- For available quantum gates, we provide the definition of some important composite gates not included in the standard QASM gate library in a "customgates.inc". Hierarchical definition for multi-controlled X gate contains 45060 lines for qubit number $n = 14$ in OpenQASM format, which is impossible for AI models to accurately generate at the time. Providing these as a .inc file guarantees the correctness of OpenQASM's grammar while avoiding the generation of complicated gates, which is a distraction from the original design task.

- To verify models' output automatically without human evaluation, we compose verification functions to validate the syntax of QASM / Qiskit and the functionality of the implemented circuits / codes. Since comprehensive Logic Equivalence Checking (LEC) might be inefficient for the throughput of LLM inference, we perform the verification by directly checking the correctness of output with extensive test cases.

Based on theses principles, we proposed the framework of QCircuitNet. Below is a more detailed explanation for the 7 components of the dataset:

1. **Problem Description:** carefully hand-crafted prompts stating the oracle to be constructed or the target problem to be solved in natural language and latex math formulas. If the problem involves the usage of a quantum oracle or composite gates beyond the standard gate library, the interfaces of the oracle / gate will also be included (input qubits, output qubits, function mechanism).

2. **Generation Code:** one general Qiskit (Javadi-Abhari et al., 2024) code to create quantum circuits for oracles or algorithms of different settings, such as distinct secret strings or various qubit numbers. We choose Qiskit as the main experiment platform because it is a general quantum programming software widely used for the complete workflow from creating quantum circuits to transpiling, simulation, and execution on real hardware.

3. **Algorithm Circuit:** a .qasm file storing the quantum circuit for each specific setting. We choose OpenQASM 3.0 (Cross et al., 2022) as the format to store the quantum circuits, because Qiskit, as a python library, can only create quantum circuits at runtime instead of explicitly saving the circuits at gate level.[1]

4. **Post-Processing Function:** this is for Algorithm Design task only, see Section 4.1.2. The function takes a complete quantum circuit as input, uses the Qiskit AerSimulator to execute the circuit, and returns the final answer to the original problem according to the simulation results. For state preparation problems such as creating a GHZ state of $n$ qubits, this function returns the qubit indices of the generated state.

5. **Oracle / Gate Definition:** a .inc file to provide definitions of composite gates or oracles. For oracle construction tasks, this only includes the definition of composite gates required to build the oracle. For algorithm design tasks, we also provide the gate definition of the oracle in this file, which successfully delivers the oracle in a black-box way.

6. **Verification Function:** a function to evaluate whether the implemented oracle / algorithm achieves the desired purpose with grammar validation and test cases verification. If there exist grammar errors, the function returns -1 and provides detailed error message, which can be used as feedback for the LLM to improve through interactive reasoning. If the program can execute successfully, the function returns a score between $[0, 1]$ indicating the success rate on test cases.[2]

7. **Dataset Creation Script:** the script to create the dataset from scratch in the format suitable for benchmarking / fine-tuning LLMs. It contains the following functions: 1. generate primitive QASM circuits. 2. extract gate definitions and add include instructions to create algorithm circuit, the direct output of model. 3. validate and verify the correctness of the data points in the dataset. 4. concatenate algorithm circuit with problem description as a json file for the benchmark pipeline.

This structure of QCircuitNet provides a general framework to formulate quantum algorithm design for large language models, with an easy extension to more advanced quantum algorithms.

---

[1]Although currently the Qiskit APIs for importing and dumping OpenQASM 3.0 files are still in experimental stage, we choose to adopt version 3.0 over 2.0 in that it supports saving parameterized circuits, which allows for extending the framework to variational quantum algorithms (Cerezo et al., 2021).

[2]The verification function explicitly integrates the oracle / gate definition library with output algorithm circuit since Qiskit importer for OpenQASM 3.0 does not support non-standard gate libraries currently.

# 5 EXPERIMENTS

## 5.1 BENCHMARKING LLMS ON QCIRCUITNET

We benchmark the quantum algorithm design capabilities of leading closed-source and open-source large language models using QCircuitNet. The workflow of our benchmark is illustrated in Figure 2. The total computation cost is approximately equivalent to two days on an A100 GPU.

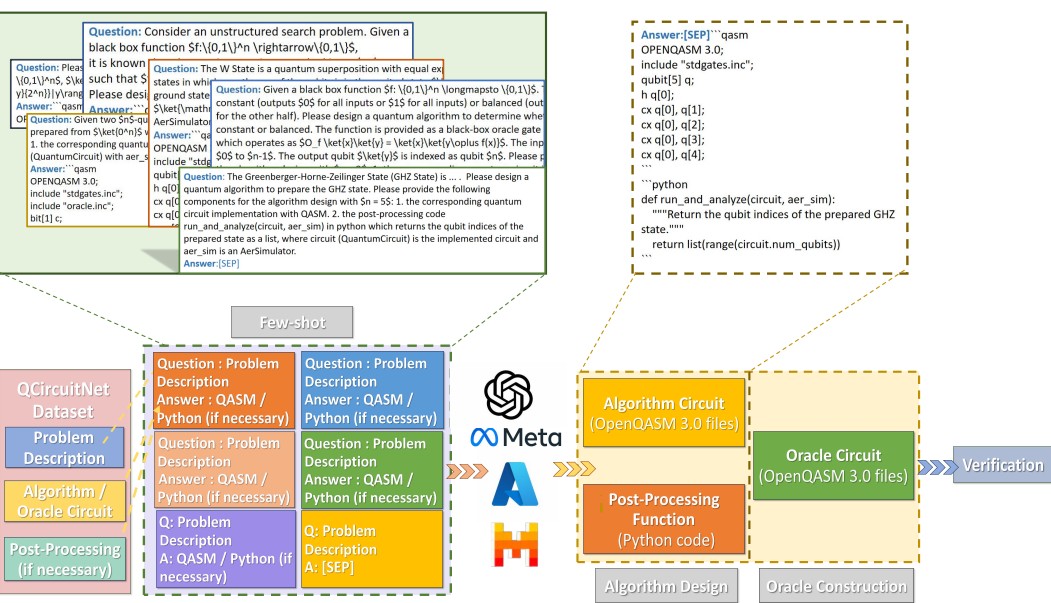

Figure 2: Flowchart of benchmarking QCircuitNet.

**Models.** Recently, the GPT series models have become the benchmark for generative models due to their exceptional performance. Specifically, we include two models from OpenAI, GPT-3.5-turbo (Brown et al., 2020) and GPT-4 (OpenAI et al., 2024), in our benchmark. Additionally, the LLAMA series models (Touvron et al., 2023a;b) are widely recognized as leading open-source models, and we have selected LLAMA-3-8B for our study. For a comprehensive evaluation, we also benchmark Phi-3-medium-128k (Abdin et al., 2024) and Mistral-7B-v0.3 (Jiang et al., 2023).

**Prompts.** We employ a few-shot learning framework, a prompting technique that has shown considerable success in generative AI (Xie et al., 2021). In this approach, we utilize either 1 or 5 examples, followed by a problem description. To ensure we do not train and test on the same quantum algorithm, we implement k-fold validation. This method involves using one problem as the test set while the remaining problems serve as the training set, rotating through each problem one at a time.

**Evaluation Metrics.** We use three evaluation metrics (see Appendix C.1 for more details):

1. BLEU Score: this metric measures how closely the generated code matches the reference code, with a higher BLEU score indicating greater similarity.

2. Verification function: this function checks the syntax validation and the result correctness of the code produced by the language model.

3. Byte Perplexity: this metric evaluates the model's ability to predict the next byte in a sequence. Lower byte perplexity indicates better performance by reflecting the model's predictive accuracy.

The results for BLEU and verification function score are shown in Figure 3, Table 1, and Table 2. We include the results of Byte Perplexity in Appendix C.2.

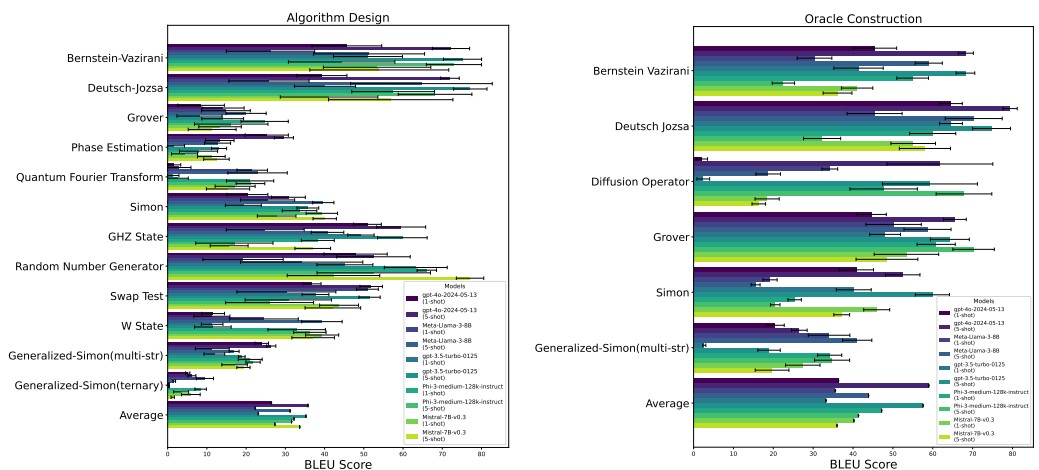

Figure 3: Benchmarking algorithm design and oracle construction in BLEU scores.

Table 1: Benchmarking algorithm design in verification function scores.

| Model | Shot | Bernstein Vazirani | Deutsch Jozsa | Grover | Phase Estimation | QFT | Simon | GHZ | Random Number Generator | Swap Test | W State | Generalized Simon (multi-str) | Generalized Simon (ternary) | Avg |
|---|---|---|---|---|---|---|---|---|---|---|---|---|---|---|
| gpt4o | 1 | -0.8462 (±0.1042) | -0.5538 (±0.1986) | -0.7089 (±0.1879) | -0.8423 (±0.0000) | -1.0000 (±0.0000) | -0.6692 (±0.1447) | -0.8462 (±0.1538) | -1.0000 (±0.0000) | -1.0000 (±0.0000) | -1.0000 (±0.0000) | -0.8844 (±0.1156) | -0.6667 (±0.3333) | -0.8348 |
| gpt4o | 5 | -0.3054 (±0.2086) | 0.0135 (±0.2070) | -0.2071 (±0.2089) | -0.5846 (±0.3107) | -0.6154 (±0.1804) | -0.3692 (±0.1443) | -0.1538 (±0.2738) | -0.4967 (±0.2210) | -0.8700 (±0.1300) | -0.9231 (±0.0769) | -0.4889 (±0.2077) | 0.0000 (±0.0000) | -0.4167 |
| Llama3 | 1 | -0.2308 (±0.2571) | -0.7692 (±0.1216) | -0.7143 (±0.1844) | -1.0000 (±0.1429) | -0.9231 (±0.0769) | -1.0000 (±0.0000) | -0.6154 (±0.1804) | -0.9285 (±0.0715) | -1.0000 (±0.0000) | -1.0000 (±0.1404) | -1.0000 (±0.0000) | -1.0000 (±0.0000) | -0.7972 |
| Llama3 | 5 | 0.0769 (±0.2107) | -0.2308 (±0.1662) | -0.5393 (±0.2185) | -0.9231 (±0.0000) | -0.7692 (±0.1662) | -0.8462 (±0.1042) | -0.3846 (±0.1404) | -0.7276 (±0.1468) | -1.0000 (±0.0000) | -0.1538 (±0.1042) | -0.8889 (±0.1111) | -1.0000 (±0.0000) | -0.6155 |
| gpt3.5 | 1 | -0.8462 (±0.1042) | -0.7154 (±0.1503) | -0.5679 (±0.2037) | -1.0000 (±0.0000) | -1.0000 (±0.0000) | -0.6231 (±0.1680) | -0.8462 (±0.1538) | -1.0000 (±0.0000) | -1.0000 (±0.0000) | -1.0000 (±0.0000) | -0.7778 (±0.1470) | -1.0000 (±0.0000) | -0.8647 |
| gpt3.5 | 5 | -0.6154 (±0.1404) | -0.0571 (±0.1934) | -0.0500 (±0.1687) | -0.9154 (±0.0000) | -0.6538 (±0.1538) | -0.1646 (±0.1395) | -0.2308 (±0.2809) | -0.4513 (±0.2410) | -0.8778 (±0.1222) | -0.8462 (±0.1042) | -0.3311 (±0.1672) | 0.0000 (±0.0000) | -0.4328 |
| Phi3 | 1 | -0.8462 (±0.1538) | -0.7750 (±0.1527) | -1.0000 (±0.0000) | -1.0000 (±0.0000) | -1.0000 (±0.0000) | -1.0000 (±0.0000) | -0.8462 (±0.2130) | -1.0000 (±0.0000) | -0.8878 (±0.1122) | -0.8462 (±0.1042) | -1.0000 (±0.0000) | -1.0000 (±0.0000) | -0.8950 |
| Phi3 | 5 | -0.6577 (±0.1891) | -0.3821 (±0.2342) | -0.8286 (±0.1714) | -0.6923 (±0.0000) | -1.0000 (±0.0000) | -0.6100 (±0.1425) | -0.9231 (±0.0769) | -0.3569 (±0.2402) | -0.8333 (±0.1667) | -0.8462 (±0.1042) | -0.8889 (±0.1111) | -1.0000 (±0.0000) | -0.7516 |
| Mistral | 1 | -0.8462 (±0.1042) | -0.8590 (±0.1410) | -0.7107 (±0.1868) | -1.0000 (±0.0000) | -1.0000 (±0.0000) | -0.9192 (±0.0808) | -0.7692 (±0.1662) | -1.0000 (±0.0000) | -1.0000 (±0.0000) | -0.6923 (±0.1332) | -0.8889 (±0.1111) | -1.0000 (±0.0000) | -0.8905 |
| Mistral | 5 | -0.6246 (±0.1664) | -0.6667 (±0.1820) | -0.4071 (±0.2106) | -1.0000 (±0.1429) | -0.9231 (±0.0769) | -0.9115 (±0.0885) | -0.6923 (±0.1332) | -0.8820 (±0.1180) | -1.0000 (±0.0000) | -0.5385 (±0.1439) | -0.8889 (±0.1111) | -0.6667 (±0.3333) | -0.7668 |

Table 2: Benchmarking oracle construction in verification function scores.

| Model | Shot | Bernstein-Vazirani | Deutsch-Jozsa | Diffusion-Operator | Grover | Simon | Generalized-Simon (multi-str) | Avg |
|---|---|---|---|---|---|---|---|---|
| gpt4o | 1 | -0.3200 (±0.0530) | -0.0100 (±0.0438) | -0.8462 (±0.1538) | -0.9885 (±0.0115) | -0.4674 (±0.0545) | -0.3750 (±0.0870) | -0.5012 |
| gpt4o | 5 | -0.1100 (±0.0399) | 0.0800 (±0.0506) | -0.3077 (±0.2083) | -0.9540 (±0.0279) | -0.0870 (±0.0295) | -0.3125 (±0.0832) | -0.2819 |
| Llama3 | 1 | -0.7300 (±0.0468) | -0.5000 (±0.0704) | -0.3846 (±0.1404) | -1.0000 (±0.0000) | -0.6848 (±0.0487) | -0.9375 (±0.0435) | -0.7061 |
| Llama3 | 5 | -0.0500 (±0.0359) | 0.1700 (±0.0551) | -0.8462 (±0.1042) | -1.0000 (±0.0000) | -0.6413 (±0.0503) | -0.6875 (±0.0832) | -0.5092 |
| gpt3.5 | 1 | -0.3500 (±0.0539) | -0.0400 (±0.0470) | -0.8462 (±0.1538) | -1.0000 (±0.0000) | -0.3696 (±0.0529) | -1.0000 (±0.0000) | -0.6010 |
| gpt3.5 | 5 | -0.1100 (±0.0373) | 0.0200 (±0.0531) | -0.3077 (±0.2083) | -0.9770 (±0.0162) | -0.1087 (±0.0326) | -0.4063 (±0.0989) | -0.3149 |
| Phi3 | 1 | -0.6800 (±0.0510) | -0.6100 (±0.0584) | -0.9231 (±0.0769) | -1.0000 (±0.0000) | -0.7500 (±0.0454) | -1.0000 (±0.0000) | -0.8272 |
| Phi3 | 5 | -0.5400 (±0.0521) | -0.4300 (±0.0685) | -1.0000 (±0.0000) | -1.0000 (±0.0000) | -0.8370 (±0.0387) | -0.9063 (±0.0524) | -0.7855 |
| Mistral | 1 | -0.4000 (±0.0512) | -0.4300 (±0.0640) | -0.9231 (±0.0769) | -0.9540 (±0.0279) | -0.6087 (±0.0512) | -0.9375 (±0.0435) | -0.7089 |
| Mistral | 5 | -0.3700 (±0.0506) | -0.1300 (±0.0734) | -1.0000 (±0.0000) | -0.9195 (±0.0373) | -0.2391 (±0.0447) | -0.9063 (±0.0524) | -0.5942 |

The results illustrate that most models achieve better scores in the five-shot setting, which indicates their ability to learn effectively from contextual examples. Notably, models perform well on tasks like Bernstein-Vazirani and Deutsch-Jozsa but struggle with more complex algorithms such as Grover, phase estimation, and quantum Fourier transform, highlighting differences in task difficulty. Furthermore, although the BLEU scores show a general trend of consistency with verification scores,

some discrepancies arise, such as the swap test showing relatively high BLEU scores but incorrect algorithm generation by most models. This observation emphasizes the need for complementary evaluation metrics beyond BLEU to accurately assess model performance, which highlights the importance of our verification function. Additionally, GPT-4o and GPT-3.5 consistently excel in long-context comprehension, significantly outperforming other models across tasks, which highlights their superior in-context learning capabilities. For more detailed analysis of the experimental results, we refer to Appendix C.1.

**Types of Errors Made by LLMs.** In Appendix C.3, we include several case studies to illustrate and analyze various types of errors made by LLMs. For example, GPT-4o tends to use advanced OpenQASM 3.0 features unsupported by Qiskit yet and novel namespace which might result in global conflicts in one-shot setting. This tendency to improvise by drawing on pre-trained knowledge rather than closely following the syntax of the example leads to avoidable "errors" and low verification scores. This issue is significantly alleviated in the 5-shot setting, highlighting GPT-4o's strong in-context learning ability.

## 5.2 FINE-TUNING ON QCIRCUITNET

Although QCircuitNet is targeted as a benchmark dataset at current stage, we consider fine-tuning / training from scratch based on our dataset as an interesting and important research direction. The unique nature of quantum data requires novel fine-tuning methods and model architecture designs, which could serve as a standalone topic. For a primitive demonstration, we present fine-tuning results on data from the oracle construction task here.

Following Dettmers et al. (2024), we quantize the model to 8-bits and then train it with LORA (Hu et al., 2022). In our experiments, we use fp16 computational datatype. We set LoRA $r = 16, \alpha = 32$ and add LoRA modules on all the query and value layers. We also use AdamW (Loshchilov & Hutter, 2019) and LoRA dropout of $0.05$. The results are shown as follows:

Table 3: Fine-tuning oracle construction scores.

| Score | Model | Setting | Bernstein-Vazirani | Deutsch-Jozsa | Grover | Simon | Clifford | Universal | Avg |
|---|---|---|---|---|---|---|---|---|---|
| BLEU | gpt4o | few-shot(5) | 95.6388 (±0.3062) | 91.0564 (±0.6650) | 92.0620 (±0.6288) | 80.3390 (±2.0900) | 39.5469 (±3.6983) | 33.3673 (±3.1007) | 72.0017 |
| | Llama3 | few-shot(5) | 53.5574 (±5.2499) | 69.8996 (±5.7812) | 61.3102 (±5.4671) | 26.3083 (±2.0048) | 13.0729 (±0.9907) | 13.4185 (±1.2299) | 39.5945 |
| | Llama3 | finetune | 76.0480 (±7.9255) | 71.8378 (±2.4179) | 67.7892 (±7.8900) | 43.8469 (±3.2998) | 10.8978 (±0.6169) | 7.1854 (±0.5009) | 46.2675 |
| Verification | gpt4o | few-shot(5) | 0.0000 (±0.0246) | 0.4300 (±0.0590) | 0.0000 (±0.1005) | -0.0200 (±0.0141) | -0.0333 (±0.0401) | -0.1023 (±0.0443) | 0.0457 |
| | Llama3 | few-shot(5) | -0.2700 (±0.0468) | 0.0900 (±0.0668) | -0.5200 (±0.0858) | -0.6600 (±0.0476) | -0.7303 (±0.0473) | -0.5056 (±0.0549) | -0.4327 |
| | Llama3 | finetune | -0.1300 (±0.0485) | -0.2000 (±0.0402) | -0.3300 (±0.0900) | -0.7400 (±0.0441) | -0.8741 (±0.0343) | -0.9342 (±0.0262) | -0.5347 |
| PPL | Llama3 | few-shot(5) | 1.1967 (±0.0028) | 1.1174 (±0.0015) | 1.1527 (±0.0021) | 1.1119 (±0.0017) | 1.4486 (±0.0054) | 1.4975 (±0.0051) | 1.2541 |
| | Llama3 | finetune | 1.0004 (±0.0002) | 1.1090 (±0.0014) | 1.0010 (±0.0006) | 1.1072 (±0.0011) | 1.2944 (±0.0053) | 1.3299 (±0.0055) | 1.1403 |

We compare the performance of Llama3-8B before and after fine-tuning with case studies. Take oracle construction of Bernstein-Vazirani Problem as an example, we observed that before fine-tuning, the model would indiscriminately apply CX gates to all qubits. After fine-tuning, it began to selectively apply CX gates to qubits with '1's in the secret string. In some cases, the positions were still counted incorrectly; however, in certain instances, the model accurately identified all the positions for applying the CX gates, which is highly impressive. This improvement significantly contributed to higher scores, suggesting that the model is starting to learn the pattern for constructing certain oracles through fine-tuning. Regarding the interesting performance decrease on Clifford and universal random circuits, we conducted additional experiments on temperature and refer to Appendix C.2 for more details.

## 6 CONCLUSIONS AND FUTURE WORK

In this paper, we propose QCircuitNet, the first comprehensive, structured universal quantum algorithm dataset and quantum circuit generation benchmark for AI models. This framework formulates quantum algorithm design from the programming language perspective and includes detailed descriptions and implementation of most established and important quantum algorithms / primitives, allowing for automatic verification methodologies. Benchmarking of QCircuitNet on up-to-date LLMs is systematically conducted. Fine-tuning results also showcase the potential of QCircuitNet as a training dataset. As shown by these benchmarking and fine-tuning results, QCircuitNet helps guide LLMs for reasoning and implementation. As we gradually extend the dataset, it will hopefully contribute to interactive quantum algorithm design.

We also highlight the challenges and opportunities as follows, ranging from scalability on the quantum algorithm side to data contamination on the AI side:

**Scalability of the Approach.** Our framework is designed to scale with increasing qubit numbers and support complex quantum algorithms as long as they are efficiently implementable with polynomial gates. The dataset generation scripts are written in a generic way which can be easily extended to arbitrarily large qubit numbers. The implementation of Generalized Simon's Problem mentioned in Section 4.1.2 showcases the compatibility of our framework with more complex algorithms. The main bottleneck for scalability in the pipeline lies in the simulation process in verification function. For simplicity of demonstration and consideration of hardware noise, all the verification functions were run with classical simulations in our experiments. But the APIs we implemented are compatible with IBM hardware and can be easily adapted to quantum computers, which reduces the cost of classical simulation. As quantum hardware becomes widespread and the capabilities of large language models continue to advance, the dataset can be easily expanded to accommodate scaling.

**Data Contamination in AI Learning.** We observe a performance separation between writing general Qiskit codes and explicit gate-level circuits in QASM. Since Qiskit provides detailed tutorial with general codes for several algorithms, this may imply a *data contamination* phenomenon where LLMs rely on memorization and retrieval rather than genuine algorithm design (see Appendix C.4 for more details). Similarly, current benchmarks for AI code generation and syntax learning may also suffer from this unseen bias. Our dataset, based on QASM files created from scratch, may help circumvent this issue and serve as a stable and fair method for benchmarking AI syntax learning.

Our work leaves several **open questions** for future investigation:

- QCircuitNet is a benchmarking dataset for LLMs. It is of general interest to extend benchmarking to training, which will help LLMs better maneuver quantum algorithm design. We have implemented advanced algorithms such as the Generalized Simon's Problem, but this in general needs implementations of more advanced algorithms to make it a more meaningful training dataset.
- Since quantum algorithms have fundamental difference from classical algorithms, novel fine-tuning methods to attempt quantum algorithm design and quantum circuit implementation, or even developments of new quantum algorithms by LLMs are solicited.
- Currently, variational quantum algorithms (Cerezo et al., 2021) can already be implemented on near-term NISQ machines (Preskill, 2018). It would be also of general interest to extend QCircuitNet to contain the design and implementation of variational quantum algorithms.

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

# A  DETAILS OF QCIRCUITNET

The QCircuitNet Dataset, along with its Croissant metadata, is available on Anonymous GitHub at the following link: https://anonymous.4open.science/r/QCircuitNet-62DF/

QCircuitNet has the following directory structure:

```
QCircuitNet
├──Oracle Construction ................... All data for the oracle construction task
│  ├──Quantum Logic Synthesis ............. Textbook-level and advanced oracles
│  ├──Problem Encoding .................... Oracles encoding application scenarios
├──Algorithm Design ................ All data for the quantum algorithm design task
│  ├──Quantum Computing ................. Universal quantum computing algorithms
│  ├──Quantum Information .............. Quantum information tasks and protocols
├──Random Circuits ................... All data for the random circuit synthesis task
   ├──Clifford ........................... Random circuits with the Clifford gate set
   ├──Universal ......................... Random circuits with the universal gate set
```

In each subdirectory, there is a folder for each specific algorithm. For instance, the folder structure for Simon's algorithm is as follows:

```
Algorithm Design
└──Quantum Computing
   └──simon ........................................ All data for the Simon's Problem
      ├──simon-dataset.py ................................. Dataset creation script
      ├──simon-generation.py ............................. Qiskit generation code
      ├──simon-post-processing.py .................... Post-processing function
      ├──simon-utils.py .......................... Utility functions for verification
      ├──simon-verification.py ............................ Verification function
      ├──simon-description.txt ............................. Problem description
      ├──simon-verification.txt ............ Verification results of the data points
      ├──full circuit ............................... Raw data of quantum circuits
      │  ├──simon-n2
      │  │  └──simon-n2-s11-k11.qasm ........... Full circuit for a concrete setting
      │  ├──simon-n3
      │  │  ├──simon-n3-s011-k001.qasm
      │  │  └──simon-n3-s011-k101.qasm
      │  └── ...
      ├──test oracle ................................... Extracted oracle definitions
      │  ├──n2
      │  │  └──trial1
      │  │     ├──oracle.inc ......................... Oracle definition as a .inc file
      │  │     └──oracle-info.txt ........ Oracle information (such as key strings)
      │  ├──n3
      │  │  ├──trial1
      │  │  │  ├──oracle.inc
      │  │  │  └──oracle-info.txt
      │  │  └──trial2
      │  │     ├──oracle.inc
      │  │     └──oracle-info.txt
      │  └── ...
      ├──simon-n2.qasm ......................... Algorithm circuit for model output
      ├──simon-n3.qasm
      └── ...
```

We expect to extend QCircuitNet under this general structure.

## A.1 FORMAT

In this subsection, we provide concrete examples to illustrate the different components of QCircuitNet. We use the case of Simon's Problem throughout the demonstration to achieve better consistency. For further details, please check the code repository.

1. **Problem Description:** this is the carefully hand-crafted description of the task in natural language and latex math formulas. The description is provided as one template for each algorithm, and the concrete settings (such as the qubit number) are replaced when creating the data points in json. The file is named as "{algorithm_name}_description.txt".

> **Problem Description Template for Simon's Problem**
>
> Given a black box function $f : \{0, 1\}^n \longmapsto \{0, 1\}^n$. The function is guaranteed to be a two-to-one mapping according to a secret string $s \in \{0, 1\}^n, s \neq 0^n$, where given $x_1 \neq x_2, f(x_1) = f(x_2) \iff x_1 \oplus x_2 = s$. Please design a quantum algorithm to find $s$. The function is provided as a black-box oracle gate named "Oracle" in the "oracle.inc" file which operates as $O_f |x\rangle |y\rangle = |x\rangle |y \oplus f(x)\rangle$. The input qubits $|x\rangle$ are indexed from 0 to $n-1$, and the output qubits $|f(x)\rangle$ are indexed from $n$ to $2n-1$. Please provide the following components for the algorithm design with $n =$ {qubit number}: 1. the corresponding quantum circuit implementation with {QASM / Qiskit}. 2. the post-processing code run_and_analyze(circuit, aer_sim) in python which simulates the circuit (QuantumCircuit) with aer_sim (AerSimulator) and returns the secret string $s$ according to the simulation results.

2. **Generation Code:** one general Qiskit code to create quantum circuits of different settings. Note that the oracle for the problem is provided as a black-box gate "oracle" here. This code is used to generate the raw data, but can also be used as a testing benchmark for writing Qiskit codes. The file is named as "{algorithm_name}_generation.py".

```python
from Qiskit import QuantumCircuit

def simon_algorithm(n, oracle):
    """Generates a Simon algorithm circuit.

    Parameters:
    - n (int): number of qubits
    - s (str): the secret string of length n

    Returns:
    - QuantumCircuit: the Simon algorithm circuit
    """
    # Create a quantum circuit on 2n qubits
    simon_circuit = QuantumCircuit(2 * n, n)

    # Initialize the first register to the |+> state
    simon_circuit.h(range(n))

    # Append the Simon's oracle
    simon_circuit.append(oracle, range(2 * n))

    # Apply a H-gate to the first register
    simon_circuit.h(range(n))

    # Measure the first register
    simon_circuit.measure(range(n), range(n))

    return simon_circuit
```

Listing 1: Qiskit generation code for Simon's algorithm.

3. **Algorithm Circuit:** the OpenQASM 3.0 format file storing the quantum circuit in gate level for each specific setting. Note that the explicit construction of "Oracle" is provided separately in "oracle.inc" file, which guarantees the usage of oracle in a black-box way. This filed is named as "{algorithm_name}_n{qubit_number}.qasm".

```
OPENQASM 3.0;
include "stdgates.inc";
include "oracle.inc";
bit[3] c;
qubit[6] q;
h q[0];
h q[1];
h q[2];
Oracle q[0], q[1], q[2], q[3], q[4], q[5];
h q[0];
h q[1];
h q[2];
c[0] = measure q[0];
c[1] = measure q[1];
c[2] = measure q[2];
```

Listing 2: OpenQASM 3.0 Code for Simon's algorithm with $n = 3$.

4. **Post-Processing Function:** this function simulates the quantum circuit and derives the final answer to the problem. The file is named as "{algorithm_name}_post_processing.py".

```
from sympy import Matrix
import numpy as np
from Qiskit import transpile

def mod2(x):
    return x.as_numer_denom()[0] % 2

def solve_equation(string_list):
    """
    A^T | I
    after the row echelon reduction, we can get the basis of the
        ↪ nullspace of A in I
    since we just need the string in binary form, so we can just use
        ↪ the basis
    if row == n-1 --> only one
    if row < n-1 --> get the first one (maybe correct or wrong)
    """
    M = Matrix(string_list).T

    # Augmented   : M | I
    M_I = Matrix(np.hstack([M, np.eye(M.shape[0], dtype=int)]))

    # RREF row echelon form , indices of the pivot columns
    # If x % 2 = 0, it will not be chosen as pivot (modulo 2)
    M_I_rref = M_I.rref(iszerofunc=lambda x: x % 2 == 0)

    # Modulo 2
    M_I_final = M_I_rref[0].applyfunc(mod2)

    # Non-Trivial solution
    if all(value == 0 for value in M_I_final[-1, : M.shape[1]]):
        result_s = "".join(str(c) for c in M_I_final[-1, M.shape[1] :])

    # Trivial solution
    else:
        result_s = "0" * M.shape[0]
```

```
        return result_s

def run_and_analyze(circuit, aer_sim):
    n = circuit.num_qubits // 2
    circ = transpile(circuit, aer_sim)
    results = aer_sim.run(circ, shots=n).result()
    counts = results.get_counts()
    equations = []
    for result, count in counts.items():
        if result != "0" * n:  # We don't use all 0 string
            y = [int(bit) for bit in result]
            equations.append(y)
    if len(equations) == 0:
        prediction = "0" * n
    else:
        prediction = solve_equation(equations)
    return prediction
```

Listing 3: Post-processing code for Simon's algorithm.

5. **Oracle / Gate Definition:** this .inc file provides the definitions of composite gates or oracles. The file is named "customgates.inc" for oracle construction tasks and "oracle.inc" for algorithm design tasks.

```
gate Oracle _gate_q_0, _gate_q_1, _gate_q_2, _gate_q_3, _gate_q_4,
    ↪ _gate_q_5 {
  cx _gate_q_0, _gate_q_3;
  cx _gate_q_1, _gate_q_4;
  cx _gate_q_2, _gate_q_5;
  cx _gate_q_2, _gate_q_5;
  x _gate_q_3;
}
```

Listing 4: One test case oracle for Simon's algorithm with $n = 3$.

For algorithm design tasks, this .inc file is accompanied with an "oracle_info.txt" file to describe the encoded information of the oracle. This helps the verification function to check the correctness of the derived answer by the model. The above test case is equipped with the following information text:

> oracle_info.txt for Simon's Problem with qubit number 3 and test case 2.
>
> Secret string: 100
> Key string: 001

6. **Verification Function:** the function to evaluate the output with grammar validation and test cases verification. The file is named as "{algorithm_name}_verification.py".

```
from simon_utils import *

def check_model(qasm_string, code_string, n):
    """Check the Simon model."""
    # Verify the syntax of the QASM code with the first test case
        ↪ oracle
    t = 1
    with open(f"test_oracle/n{n}/trial{t}/oracle.inc", "r") as file:
        oracle_def = file.read()
    full_qasm = plug_in_oracle(qasm_string, oracle_def)
    circuit = verify_qasm_syntax(full_qasm)
    if circuit is None:
        return -1
    try:
```

```
        exec(code_string, globals())
        aer_sim = AerSimulator()
        total_success = 0
        total_fail = 0
        t_range = min(10, 4 ** (n - 2))
        shots = 10
        for t in range(1, 1 + t_range):
            print(f"    Running Test Case {t}")
            with open(f"test_oracle/n{n}/trial{t}/oracle.inc", "r") as
                ↪ file:
                oracle_def = file.read()
            full_qasm = plug_in_oracle(qasm_string, oracle_def)
            circuit = loads(full_qasm)
            with open(f"test_oracle/n{n}/trial{t}/oracle_info.txt", "r"
                ↪ ) as file:
                content = file.read()
            match = re.search(r"Secret string: ([01]+)", content)
            if match:
                secret_string = match.group(1)
            else:
                raise ValueError("Secret string not found in the file."
                    ↪ )

            cnt_success = 0
            cnt_fail = 0
            for shot in range(shots):
                prediction = run_and_analyze(circuit.copy(), aer_sim)
                if not isinstance(prediction, str):
                    raise TypeError("Predicted secret string should be
                        ↪ a string.")
                if prediction == secret_string:
                    cnt_success += 1
                else:
                    cnt_fail += 1
            print(f"        Success: {cnt_success}/{shots}, Fail: {
                ↪ cnt_fail}/{shots}")
            total_success += cnt_success
            total_fail += cnt_fail
        print(f"Total Success: {total_success}; Total Fail: {total_fail
            ↪ }")
        return total_success / (total_fail + total_success)

    except Exception as e:
        print(f"Error: {e}")
        return -1
```

Listing 5: Verification function for Simon's algorithm.

This verification function is accompanied with an "{algorithm_name}_utils.py" file to provide necessary utility functions.

```
from Qiskit.qasm3 import loads
from Qiskit_aer import AerSimulator
import re

def print_and_save(message, text):
    print(message)
    text.append(message)

def plug_in_oracle(qasm_code, oracle_def):
    """Plug-in the oracle definition into the QASM code."""
    oracle_pos = qasm_code.find('include "oracle.inc";')
    if oracle_pos == -1:
```

```
raise ValueError("Oracle include statement not found in the
    ↪ file")
full_qasm = (
    qasm_code[:oracle_pos]
    + oracle_def
    + qasm_code[oracle_pos + len('include "oracle.inc";') :]
)
return full_qasm

def verify_qasm_syntax(output):
    """Verify the syntax of the output and return the corresponding
        ↪ QuantumCircuit (if it is valid)."""
    assert isinstance(output, str)
    try:
        # Parse the OpenQASM 3.0 code
        circuit = loads(output)
        print(
            "    The OpenQASM 3.0 code is valid and has been
                ↪ successfully loaded as a QuantumCircuit."
        )
        return circuit
    except Exception as e:
        print(f"    Error: The OpenQASM 3.0 code is not valid. Details:
            ↪  {e}")
        return None
```

Listing 6: Utility functions for verification of Simon's algorithm.

7. **Dataset Creation Script:** this script involves all the code necessary to create the data points from scratch. The file is named as "{algorithm_name}_dataset.py". The main function looks like this:

```
def main():
    parser = argparse.ArgumentParser()
    parser.add_argument(
        "-f",
        "--func",
        choices=["qasm", "json", "gate", "check"],
        help="The function to call: generate qasm circuit, json dataset
            ↪  or extract gate definition.",
    )
    args = parser.parse_args()
    if args.func == "qasm":
        generate_circuit_qasm()
    elif args.func == "json":
        generate_dataset_json()
    elif args.func == "gate":
        extract_gate_definition()
    elif args.func == "check":
        check_dataset()
```

Listing 7: Main function of the dataset script for Simon's algorithm.

Here the "generate_circuit_qasm()" function generates the raw data of quantum circuits in Open-QASM 3.0 format where the algorithm circuit and the oracle definition are blended, then "extract_gate_definition()" function extracts the definition of oracles and formulates the algorithm circuits into the format suitable for model output. The "check_dataset()" function is used to check the correctness of the created data points and "generate_dataset_json()" function to combine the data into json format for easy integration with the benchmarking pipeline.

## A.2 DISCUSSION OF MORE TASKS

**Problem Encoding.** In Section 4.1.1, we mentioned another category of oracle construction tasks referred to as "Problem Encoding", which involves applying quantum algorithms, such as Grover's

algorithm, to solve practical problems such as SAT and triangle finding. The crux of this process is encoding the problem constraints into Grover's oracle, thereby making this a type of oracle construction task. Unlike quantum logic synthesis, which encodes an explicit function $f(x)$ as a unitary operator $U_f$, this task involves converting the constraints of a particular problem into the required oracle form. We provide implementations of several concrete problems in this directory as demonstrations and will include more applications in future work.

**Quantum Information Protocols.** In the "Quantum Information" section of the "Algorithm Design" task, we have also implemented three important quantum information protocols: Quantum Teleportation, Superdense Coding, and Quantum Key Distribution (BB84). A brief introduction to these protocols can be found in Appendix B. We did not include the experiments for these protocols as they involve communication between two parties, which is challenging to characterize with a single OpenQASM 3.0 file. We recommend revising the post-processing function as a general classical function to schedule the communication and processing between different parties specifically for these protocols. The fundamental quantum circuits and processing codes are provided in the repository.

A.3 DATASHEET

Here we present a datasheet for the documentation of QCircuitNet.

**Motivation**

- *For what purpose was the dataset created?* It was created as a benchmark for the capability of designing and implementing quantum algorithms for LLMs.
- *Who created the dataset (e.g., which team, research group) and on behalf of which entity (e.g., company, institution, organization)?* The authors of this paper.
- *Who funded the creation of the dataset?* We will reveal the funding resources in the Acknowledgement section of the final version.

**Composition**

- *What do the instances that comprise the dataset represent (e.g., documents, photos, people, countries)?* The dataset comprises problem description, generation code, algorithm circuit, post-processing function, oracle / gate definition, verification function, and dataset creation script for various quantum algorithms.
- *How many instances are there in total (of each type, if appropriate)?* The dataset has 5 algorithms for oracle construction task and 10 algorithms for algorithm design task used for experiments. There are 3 quantum information protocols and additional problem encoding tasks not included for experiments.
- *Does the dataset contain all possible instances or is it a sample (not necessarily random) of instances from a larger set?* The dataset contains instances with restricted qubit numbers due to the current scale of real quantum hardware.
- *What data does each instance consist of?* Qiskit codes, OpenQASM 3.0 codes, python scripts, and necessary text information.
- *Are relationships between individual instances made explicit?* Yes, the way to create different instances are clearly described in Appendix A.1.
- *Are there recommended data splits?* Yes, we recommend splitting the data according to different algorithms in algorithm design task.
- *Are there any errors, sources of noise, or redundancies in the dataset?* There might be some small issues due to the dumping process of Qiskit and programming mistakes (if any).
- *Is the dataset self-contained, or does it link to or otherwise rely on external resources (e.g., websites, tweets, other datasets)?* The dataset is self-contained.
- *Does the dataset contain data that might be considered confidential (e.g., data that is protected by legal privilege or by doctor-patient confidentiality, data that includes the content of individuals' non-public communications)?* No.

- *Does the dataset contain data that, if viewed directly, might be offensive, insulting, threatening, or might otherwise cause anxiety?* No.

**Collection Process**

- *How was the data associated with each instance acquired?* The data is created by first composing Qiskit codes for each algorithm and then converting to OpenQASM 3.0 files using Qiskit.qasm3.dump function, with additional processing procedure.

- *What mechanisms or procedures were used to collect the data (e.g., hardware apparatuses or sensors, manual human curation, software programs, software APIs)?* Manual human programming and Qiskit APIs.

- *Who was involved in the data collection process (e.g., students, crowd workers, contractors), and how were they compensated (e.g., how much were crowd workers paid)?* Nobody other than the authors of the paper.

- *Over what timeframe was the data collected?* The submitted version of the dataset was created in May and June 2024.

**Uses**

- *Has the dataset been used for any tasks already?* It has been used in this paper to benchmark LLM's ability for quantum algorithm design.

- *Is there a repository that links to any or all papers or systems that use the dataset?* The only paper which uses the dataset for now is this paper.

**Distribution**

- *Will the dataset be distributed to third parties outside of the entity (e.g., company, institution, organization) on behalf of which the dataset was created?* Yes, the dataset will be made publicly available on the Internet after the review process.

- *How will the dataset be distributed (e.g., tarball on website, API, GitHub)?* It will be distributed on the GitHub platform.

- *Will the dataset be distributed under a copyright or other intellectual property (IP) license, and/or under applicable terms of use (ToU)?* The dataset is distributed under CC BY 4.0.

- *Have any third parties imposed IP-based or other restrictions on the data associated with the instances?* No.

- *Do any export controls or other regulatory restrictions apply to the dataset or to individual instances?* No.

**Maintenance**

- *Who will be supporting/hosting/maintaining the dataset?* The authors of this paper.

- *How can the owner/curator/manager of the dataset be contacted (e.g., email address)?* The email for contact will be provided after the review process.

- *Is there an erratum?* Not at this time.

- *Will the dataset be updated (e.g., to correct labeling errors, add new instances, delete instances)?* Yes, it will be continually updated.

- *If others want to extend/augment/build on/contribute to the dataset, is there a mechanism for them to do so?* Yes, they can do so with the GitHub platform.

A.4 COPYRIGHT AND LICENSING TERMS

This work is distributed under a CC BY 4.0 license. The implementation of the code references open-source projects such as Qiskit, QuantumKatas, Cirq, and NWQBench. We bear responsibility in case of violation of rights.

## B  ADDITIONAL PRELIMINARIES FOR QUANTUM COMPUTING AND QUANTUM INFORMATION

**Quantum Circuit Diagram.**    A quantum algorithm is composed of a series of quantum gates. By default, a quantum algorithm starts from the all-0 state $|0^n\rangle$. A quantum algorithm can be illustrated by its quantum gate diagram, drawn from left to right. The initial all-0 state is placed at the left side of the diagram. After that, whenever we apply a quantum gate, it is placed on the corresponding qubits, from left to right. At the end of the quantum gates, we need to measure and read the outputs, and these measurements are placed at the right side of the diagram. See Figure 4 for the quantum gate diagram of Simon's algorithm (Simon, 1997).

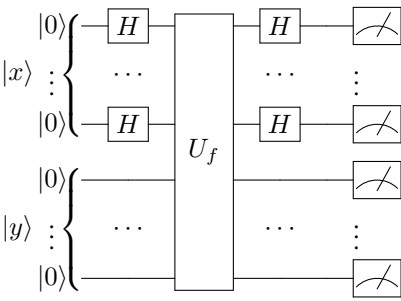

Figure 4: Quantum gate diagram of Simon's algorithm.

**Superdense Coding.**    Superdense coding (Bennett & Wiesner, 1992) is a quantum communication protocol that allows Alice to transmit two classical bits of information to Bob by sending only one qubit, given that they share a pair of entangled qubits. The protocol can be divided into five steps:

1.  **Preparation:** Charlie prepares a maximally entangled Bell state, such as $|\beta_{00}\rangle = \frac{1}{\sqrt{2}}(|00\rangle + |11\rangle)$.

2.  **Sharing:** Charlie sends the qubit 1 to Alice and the qubit 2 to Bob. Alice and Bob can be separated by an arbitrary distance.

3.  **Encoding:** Depending on the two classical bits $zx \in \{00, 01, 10, 11\}$ that Alice wants to send, she applies the corresponding quantum gate operation to her qubit, transforming the Bell state $|\beta_{00}\rangle$ into one of the four Bell states:

$$|\beta_{00}\rangle = \frac{1}{\sqrt{2}}(|00\rangle + |11\rangle) \text{ if } zx = 00$$

$$|\beta_{01}\rangle = \frac{1}{\sqrt{2}}(|01\rangle + |10\rangle) \text{ if } zx = 01$$

$$|\beta_{10}\rangle = \frac{1}{\sqrt{2}}(|00\rangle - |11\rangle) \text{ if } zx = 10$$

$$|\beta_{11}\rangle = \frac{1}{\sqrt{2}}(|01\rangle - |10\rangle) \text{ if } zx = 11$$

Alice achieves these transformations by applying the operation $Z^z X^x$ to her qubit, where $Z$ is the phase-flip gate, $X$ is the bit-flip gate. Specifically:

- If $zx = 00$, Alice applies $Z^0 X^0 = I$ (identity gate).
- If $zx = 01$, Alice applies $Z^0 X^1 = X$ (bit-flip gate).
- If $zx = 10$, Alice applies $Z^1 X^0 = Z$ (phase-flip gate).
- If $zx = 11$, Alice applies $Z^1 X^1 = ZX = iY$ gate.

4.  **Sending:** Alice sends her qubit to Bob through a quantum channel.

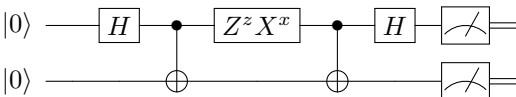

Figure 5: Quantum circuit diagram for superdense coding.

5. **Decoding:** Bob applies a CNOT gate followed by a Hadamard gate to the two qubits, transforming the entangled state into the corresponding computational basis state $|zx\rangle$. By measuring the qubits, Bob obtains the two classical bits $zx$ sent by Alice.

Superdense coding exploits the properties of quantum entanglement to transmit two classical bits of information using only one qubit. The quantum circuit diagram for superdense coding is shown in Figure 5.

**Quantum Teleportation.** Quantum teleportation (Bennett et al., 1993) is a technique for transferring quantum information from a sender (Alice) to a receiver (Bob) using shared entanglement and classical communication. The protocol can be described as follows:

1. **Preparation:** Telamon prepares a maximally entangled Bell state, such as $|\beta_{00}\rangle = \frac{1}{\sqrt{2}}(|00\rangle + |11\rangle)$.

2. **Sharing:** Alice has qubit 1 in the state $|\psi\rangle = \alpha|0\rangle + \beta|1\rangle$, which she wants to teleport to Bob. Telamon shares qubit 2 with Alice and qubit 3 with Bob, creating the shared entangled state $|\beta_{00}\rangle_{23}$.

3. **Encoding:** Alice wants to teleport an unknown quantum state $|\psi\rangle = \alpha|0\rangle + \beta|1\rangle$ to Bob. She applies a CNOT gate to qubits 1 and 2, with qubit 1 as the control and qubit 2 as the target. Then, she applies a Hadamard gate to qubit 1. The resulting state of the three-qubit system is:

$$|\Psi\rangle = \frac{1}{2}[|\beta_{00}\rangle(\alpha|0\rangle + \beta|1\rangle) + |\beta_{01}\rangle(\alpha|1\rangle + \beta|0\rangle)$$
$$+ |\beta_{10}\rangle(\alpha|0\rangle - \beta|1\rangle) + |\beta_{11}\rangle(\alpha|1\rangle - \beta|0\rangle)].$$

4. **Measurement:** Alice measures qubits 1 and 2 in the Bell basis and obtains one of four possible outcomes: $|\beta_{00}\rangle$, $|\beta_{01}\rangle$, $|\beta_{10}\rangle$, or $|\beta_{11}\rangle$. This measurement collapses the three-qubit state into one of the following:

$$|\beta_{00}\rangle \otimes (\alpha|0\rangle + \beta|1\rangle)$$
$$|\beta_{01}\rangle \otimes (\alpha|1\rangle + \beta|0\rangle)$$
$$|\beta_{10}\rangle \otimes (\alpha|0\rangle - \beta|1\rangle)$$
$$|\beta_{11}\rangle \otimes (\alpha|1\rangle - \beta|0\rangle)$$

5. **Classical Communication:** Alice sends the result of her measurement (two classical bits) to Bob via a classical channel.

6. **Reconstruction:** Depending on the classical information received from Alice, Bob applies the operation $Z^z X^x$ to qubit 3, where $z$ and $x$ correspond to the two classical bits sent by Alice:

   - If Alice measured $|\beta_{00}\rangle$, she sends $zx = 00$, and Bob applies $Z^0 X^0 = I$ (identity operation).
   - If Alice measured $|\beta_{01}\rangle$, she sends $zx = 01$, and Bob applies $Z^0 X^1 = X$ (bit-flip).
   - If Alice measured $|\beta_{10}\rangle$, she sends $zx = 10$, and Bob applies $Z^1 X^0 = Z$ (phase-flip).
   - If Alice measured $|\beta_{11}\rangle$, she sends $zx = 11$, and Bob applies $Z^1 X^1 = ZX = iY$ (bit-flip and phase-flip).

   After applying the appropriate operation, Bob's qubit 3 will be in the state $|\psi\rangle = \alpha|0\rangle + \beta|1\rangle$, which is the original state that Alice wanted to teleport.

The quantum circuit diagram for quantum teleportation is shown in Figure 6.

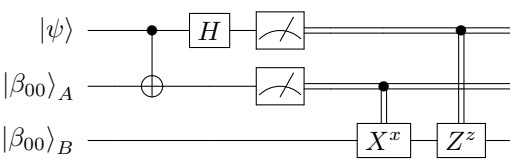

Figure 6: Quantum circuit diagram for quantum teleportation

**Quantum Key Distribution.** Quantum key distribution (QKD) (Bennett & Brassard, 1984) is a secure communication protocol that allows two parties, Alice and Bob, to produce a shared random secret key, which can then be used to encrypt and decrypt messages. The security of QKD is based on the fundamental principles of quantum mechanics that measuring a qubit can change its state. One of the most well-known QKD protocols is the BB84 protocol, which works as follows:

1. Alice randomly generates a bit string and chooses a random basis (X or Z) for each bit. She then encodes the bits into qubits using the chosen bases and sends them to Bob through a quantum channel.

2. Bob measures the received qubits in randomly chosen bases (X or Z) and records the results.

3. Alice and Bob communicate over a public classical channel to compare their basis choices. They keep only the bits for which their basis choices coincide and discard the rest.

4. Alice and Bob randomly select a subset of the remaining bits and compare their values. If the error rate is below a certain threshold, they conclude that no eavesdropping has occurred, and the remaining bits can be used as a secret key. If the error rate is too high, they abort the protocol, as it indicates the presence of an eavesdropper (Eve).

The security of the BB84 protocol relies on the fact that any attempt by Eve to measure the qubits during transmission will introduce detectable errors, alerting Alice and Bob to the presence of an eavesdropper.

## C  ADDITIONAL EXPERIMENT RESULTS

In this section, we include detailed analysis of the experiments and additional experiment results. In Section C.1, we introduce the metrics: BLEU score, verification score, and byte perplexity, and provide a detailed analysis for the experiments on BLEU and verification score. In Section C.2, we include additional experiments on perplexity score and temperatures. In Section C.3, we present concrete cases of typical patterns observed in model outputs. In Section C.4, we analyze the data contamination phenomenon revealed by our benchmark.

### C.1  METRICS

**BLEU Score.** Bilingual Evaluation Understudy (BLEU) score is a metric used to evaluate the quality of machine-translated text compared to human-translated text. It measures how close the machine translation is to one or more reference translations. The BLEU score evaluates the quality of text generated by comparing it with one or more reference texts. It does this by calculating the n-gram precision, which means it looks at the overlap of n-grams (contiguous sequences of n words) between the generated text and the reference text. Originally the BLEU score ranges from 0 to 1, where 1 indicates a perfect match with the reference translations. Here rescaling the score makes it ranges from 0 to 100.

The BLEU score, originally designed for machine translation, can also be effectively used for evaluating algorithm generation tasks. Just as BLEU measures the similarity between machine-translated text and human reference translations, it can measure the similarity between a generated algorithm and a gold-standard algorithm. This involves comparing sequences of tokens to assess how closely the generated output matches the reference solution. In the context of algorithm generation, n-grams can represent sequences of tokens or operations in the code. BLEU score captures the precision

of these n-grams, ensuring that the generated code aligns closely with the expected sequences found in the reference implementation.

The formula for BLEU score is given by:

$$\text{BLEU} = BP \cdot \exp\left(\sum_{n=1}^{N} w_n \log p_n\right).$$

where $BP$ is the acronym for brevity penalty, $w_n$ is the weight for the n-gram precision (typically $\frac{1}{N}$ for uniform weights), $p_n$ is the precision for n-grams. BP is calculated as:

$$BP = \begin{cases} 1 & \text{if } c > r \\ e^{1-\frac{r}{c}} & \text{if } c \leq r \end{cases}.$$

where $c$ is the length of the generated text and $r$ is the length of the reference text. Furthermore, n-gram precision $p_n$ is calculated as:

$$p_n = \frac{\sum_{C \in \text{Candidates}} \sum_{n-\text{gram} \in C} \min(\text{Count}(n - \text{gram in candidate}), \text{Count}(n - \text{gram in references}))}{\sum_{C \in \text{Candidates}} \sum_{n-\text{gram} \in C} \text{Count}(n - \text{gram in candidate})}.$$

This formulation ensures that the BLEU score takes into account both the precision of the generated n-grams and the overall length of the translation, providing a balanced evaluation metric.

In our experiments, the BLEU scores for various quantum algorithm design tasks are illustrated in Figure 3(a). This figure not only displays the average performance of each model but also highlights the differences in performance across individual quantum algorithm tasks. The first notable observation is that the figure clearly demonstrates the varying levels of difficulty among quantum algorithms. For example, models achieve higher BLEU scores on tasks such as Bernstein-Vazirani and Deutsch-Jozsa, whereas they perform significantly worse on tasks like Grover, phase estimation, and quantum Fourier transform. This indicates that the former tasks are considerably easier than the latter ones. Another significant observation is that most models score higher in a five-shot prompt compared to a one-shot prompt, which confirms the large language models' ability to improve performance through contextual learning.

Similar patterns are observed in oracle construction tasks, as illustrated in Figure 3(b). The figure highlights that the Diffusion Operator task is notably more challenging than the Grover oracle construction task. Interestingly, we found that adding more in-context examples actually reduced the performance of the Phi-3-medium-128k-instruct and Mistral-7B-v0.3 models. This decline in performance could be attributed to the significant differences between each oracle construction task, which may be too out-of-distribution. Consequently, the additional examples might cause the models to overfit to the specific examples provided in the context, rather than generalizing well across different tasks.

**Detailed Analysis of Verification Score.** In addition to evaluating the BLEU score, we conducted an experiment to measure the correctness of the machine-generated algorithms, and the results are shown in Table 1. By running a verification function, we discovered that phase estimation and the swap test are significantly more challenging than other problems, leading most models to score -1 (indicating they cannot even generate the correct syntax). Notably, the BLEU score for the swap test is above average compared to other algorithms, yet almost none of the models produced a correct algorithm. This discrepancy highlights a critical limitation of using BLEU as a metric for algorithm evaluation. BLEU measures average similarity, but even a single mistake in an algorithm can render it entirely incorrect, thus failing to capture the true accuracy and functionality of the generated algorithms. Another important finding is that in a five-shot setting, GPT-4 and GPT-3.5 surpass all other models by a large margin. This demonstrates their exceptional capabilities, particularly in long-context comprehension and in-context learning. These models not only excel in understanding and generating text based on minimal examples but also maintain high performance over extended sequences, highlighting their advanced architecture and training methodologies.

The verification results of the oracle construction task, as shown in Table 2, confirm our previous conclusions. In the five-shot setting, GPT-4 and GPT-3.5 consistently outperform all other models.

Additionally, this table highlights the inconsistency between BLEU scores and verification scores. For instance, while the Diffusion Operator task achieves the lowest BLEU score, it is the Grover oracle construction that receives the lowest verification score. This discrepancy suggests that BLEU scores may not fully capture the performance of models in certain complex tasks, and it is necessary to include verification score as a comprehensive evaluation.

**Byte Perplexity.** Perplexity is a measure of how well a probability distribution or a probabilistic model predicts a sample. In the context of language models, it quantifies the uncertainty of the model when it comes to predicting the next element in a sequence. Byte perplexity specifically deals with sequences of bytes, which are the raw binary data units used in computer systems. For our purposes, we consider byte perplexity under UTF-8 encoding, a widely used character encoding standard that represents each character as one or more bytes.

For a given language model, let $p(x_i|x_{<i})$ be the probability of the $i$-th byte $x_i$ given the preceding bytes $x_{<i}$. If we have a sequence of bytes $x = (x_1, x_2, \ldots, x_N)$, the perplexity $\text{PPL}(x)$ of the model on this sequence is defined as:

$$\text{PPL}(x) = 2^{-\frac{1}{N} \sum_{i=1}^{N} \log_2 p(x_i|x_{<i})}.$$

A notable feature of byte perplexity is that, it does not rely on any specific tokenizer, making it versatile for comparing different models. Therefore, byte perplexity can be used to measure the performance in quantum algorithm generation tasks. In such tasks, a lower byte perplexity indicates a better-performing model, as it means the model is more confident in its predictions of the next byte in the sequence.

## C.2 Additional Experiments on PPL Score and Temperatures

The Byte Perplexity results, shown in Figure 7, provide valuable insights into the performance of our model. Evaluated in a zero-shot setting, byte perplexity trends closely mirror those observed with BLEU scores. This alignment suggests that our model's predictive capabilities are consistent across Perplexity and BLEU evaluation metrics. Specifically, in the context of quantum algorithm design tasks, the results indicate that the Bernstein-Vazirani and Deutsch-Jozsa algorithms are relatively straightforward for the model, whereas the Simon algorithm presents greater difficulty. This differentiation highlights the varying levels of complexity inherent in these quantum algorithms.

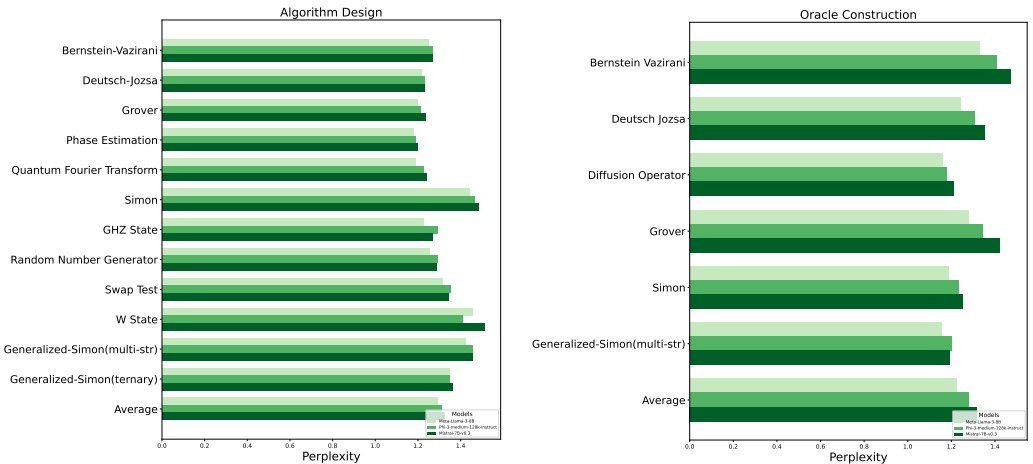

Figure 7: Benchmarking algorithm design and oracle construction in perplexity scores.

Regarding the counter-intuitive phenomenon where the performance on Clifford and universal random circuits decreases after fine-tuning, we conducted additional experiments and fine-tuned the model on 4,800 samples specifically for the Clifford task. Upon closer inspection, we observed that the model more frequently generated outputs with infinite loops and increased monotony, often producing

repetitive gate patterns and repeatedly cycling over the same qubit after fine-tuning. We further conducted experiments with different "temperature" parameters, which control the randomness of predictions. Formally, let $T > 0$ be the temperature, $z_i$ be the raw score for token $i$, the probability for token $i$ is computed as $p_i = \frac{e^{z_i/T}}{\sum_j e^{z_j/T}}$. Typically, lower temperatures make the model more conservative, while higher temperatures flatten the distribution, increasing the likelihood of generating originally less probable sequences. The results are shown in Table 4:

Table 4: Clifford Model Fine-Tuning Results Across Different Temperature Settings

| Model | Setting | Temperature | BLEU | Verification |
|---|---|---|---|---|
| Llama3 | few-shot(5) | 0 | 13.3796(±0.9508) | -0.6582(±0.0360) |
| | | 0.2 | 12.5688(±0.8276) | -0.6526(±0.0372) |
| | | 1 | 53.0431(±3.8422) | -0.1914(±0.0361) |
| Llama | finetune | 0 | 7.6261(±0.3433) | -0.8895(±0.0247) |
| | | 0.2 | 13.8714(±0.6536) | -0.7873(±0.0306) |
| | | 1 | 32.5241(±2.0548) | -0.2072(±0.0358) |

One possible explanation for this counter-intuitive result lies in the challenge of encoding quantum state vectors within a language model. In the problem description, the target quantum state is represented by a complex vector with four decimal places of precision, where the dimension scales as with the number of qubits . It is a well-known fact that LLMs generally struggle with very long floating-point numbers, which might contribute to the observed performance decline.

Another potential reason could be overfitting during fine-tuning, particularly for tasks that require high output diversity. The varying degrees of intrinsic difficulty and the amount of relevant pre-training knowledge across different tasks likely played a role. Oracle constructions are relatively simple for the model to learn. For example, in the Bernstein-Vazirani algorithm, the model only needs to apply a CNOT gate at positions corresponding to '1' bits. In contrast, the random circuits in the Clifford and Universal tasks involve more general and complex quantum state transformations, making them significantly more challenging. These tasks are also less common during pre-training, which could have hindered the model's ability to generalize without overfitting. This challenge is one of the reasons we initially considered a few-shot learning approach to be suitable.

While these are plausible hypotheses, we acknowledge that further investigation is required to draw definitive conclusions. We consider this an intriguing topic that warrants additional research.

## C.3 CASE STUDY

After carefully examining the model's output, we observed several interesting patterns. We present a series of case studies to illustrate these observations and provide possible explanations.

**Low Score for GPT-4o in One-Shot Setting.** At first glance, it is surprising that GPT-4o performs poorly on many quantum algorithms in the algorithm design task in the one-shot setting compared to Llama3-8B. Given that Llama3-8B has a relatively smaller parameter scale, the results should have been the other way around. A closer examination of the model's output reveals the potential reason: while Llama3-8B closely mimics the input examples, GPT-4o tends to improvise, resulting in outputs that are not well captured by the current syntax support. Here are several concrete examples.

This is the OpenQASM 3.0 code output for the W state with $n = 7$. In this code, GPT-4o uses the advanced "for" loop syntax newly introduced in OpenQASM 3.0 to create the circuit. Although the code fails to produce the W state, it is syntactically correct. However, the Qiskit.qasm3 import module, which converts OpenQASM 3.0 files to QuantumCircuit objects and is used in our verification function to check the correctness of the syntax of output OpenQASM codes, is still in the experimental stage and does not support many of OpenQASM 3.0's advanced features. As a result, GPT-4o's use of these features causes the code to fail syntax validation, resulting in a score of -1.

```
OPENQASM 3.0;
include "stdgates.inc";
qubit[7] q;
h q[0];
```

```
for i in[1:6] {
    cx q[i-1], q[i];
}
```

Listing 8: OpenQASM 3.0 Code output by GPT-4o for W state with $n = 7$.

Here is another example where GPT-4o decides to assign novel names to its qubit registers, leading to a conflict in the symbol table in Scope.GLOBAL. If we substitute all the registers $x$, $y$, and $s$ with new names, the code can pass syntax validation successfully and is close to the correct solution.

```
OPENQASM 3.0;
include "stdgates.inc";
include "oracle.inc";
bit[9] s;
qubit[10] x;
qubit[11] y;
h x[0];
h x[1];
h x[2];
h x[3];
h x[4];
h x[5];
h x[6];
h x[7];
h x[8];
Oracle x[0], x[1], x[2], x[3], x[4], x[5], x[6], x[7], x[8], y;
h x[0];
h x[1];
h x[2];
h x[3];
h x[4];
h x[5];
h x[6];
h x[7];
h x[8];
measure x[0] -> s[0];
measure x[1] -> s[1];
measure x[2] -> s[2];
measure x[3] -> s[3];
measure x[4] -> s[4];
measure x[5] -> s[5];
measure x[6] -> s[6];
measure x[7] -> s[7];
measure x[8] -> s[8];
```

Listing 9: OpenQASM 3.0 Code output by GPT-4o for Bernstein-Vazirani Problem with $n = 9$.

Since Llama3-8B tends to follow the provided example more closely, it avoids such kind of errors. These factors may help explain why GPT-4o performs poorly in the one-shot setting compared to Llama3-8B.

**High Score for GPT-4o in Five-Shot Setting.** In contrast to its relatively low score in the one-shot setting, GPT-4o achieves the best overall performance in the five-shot setting for both oracle construction and algorithm design tasks.

Unlike in the one-shot setting, where GPT-4o frequently uses advanced OpenQASM 3.0 features, it learns from the examples in context and uses simpler syntax in the five-shot setting. For the W state task, here is an output in the five-shot setting:

```
OPENQASM 3.0;
include "stdgates.inc";
qubit[3] q;
h q[0];
cx q[0], q[1];
```

```
h q[1];
cx q[1], q[2];
h q[2];
```

Listing 10: OpenQASM 3.0 Code output by GPT-4o for W state with $n = 3$.

This output avoids the use of "for" loops and successfully passes the syntax validation test, although it still cannot generate the W state correctly.

In addition to adapting to plain syntax through in-context learning, GPT-4o achieves outstanding performance on more complicated tasks such as phase estimation. Here is the model output of GPT-4o on the phase estimation task with qubit number $n = 2$.

```
OPENQASM 3.0;
include "stdgates.inc";
include "oracle.inc";
bit[2] c;
qubit[2] q;
Psi q[0];
h q[1];
CU_0 q[0], q[1];
h q[1];
c[0] = measure q[1];
c[1] = measure q[0];
```

Listing 11: OpenQASM 3.0 Code output by GPT-4o for Phase Estimation with $n = 2$.

```
from Qiskit import transpile

def run_and_analyze(circuit, aer_sim):
    """Run the circuit and analyze the result."""
    circ = transpile(circuit, aer_sim)
    result = aer_sim.run(circ, shots=1).result()
    counts = result.get_counts()
    phase_str = list(counts.keys())[0]
    phase = int(phase_str, 2) / 2**2
    return phase
```

Listing 12: Post-processing code output by GPT-4o for Phase Estimation with $n = 2$.

This suite of OpenQASM 3.0 circuits and post-processing functions successfully outputs the phase within the required precision for the test case, resulting in an impressive verification score of 1.0. Despite the small number of qubits and differences from the reference implementation, the accuracy achieved is noteworthy.

These phenomena reflect that GPT-4o has impressive in-context learning abilities and overall better capabilities in designing and implementing quantum algorithms.

C.4 DATA CONTAMINATION

In our experiments, we observed an interesting phenomenon of performance discrepancy between implementing quantum algorithms with Qiskit codes for the general setting and with OpenQASM codes for a concrete case.

We provided the following prompt:

> **Prompt for Bernstein-Vazirani Problem with Qiskit**
>
> Please write the code with Qiskit implementing Bernstein-Vazirani Algorithm with qubit number n = 5.

The model answered with the following text and codes:

---

**Text Answer by GPT4o**

Here's a Python code using Qiskit to implement the Bernstein-Vazirani algorithm with $n = 5$ qubits. The secret string used in this example is $s = 10101$ (you can change this to any string of your choice).

```python
from qiskit import QuantumCircuit, Aer, transpile, assemble
from qiskit.visualization import plot_histogram

# Secret string s for the Bernstein-Vazirani problem
s = '10101'

# Number of qubits
n = len(s)

# Create a Quantum Circuit with n+1 qubits (one ancillary qubit)
bv_circuit = QuantumCircuit(n+1, n)

# Apply Hadamard gates to all qubits except the ancillary qubit
bv_circuit.h(range(n))

# Apply X and Hadamard gate to the ancillary qubit
bv_circuit.x(n)
bv_circuit.h(n)

# Construct the Oracle
for i, bit in enumerate(s):
    if bit == '1':
        bv_circuit.cx(i, n)

# Apply Hadamard gates to the first n qubits
bv_circuit.h(range(n))
# Measure the first n qubits
bv_circuit.measure(range(n), range(n))
# Visualize the circuit
bv_circuit.draw('mpl')

# Run the circuit using Qiskit Aer simulator
simulator = Aer.get_backend('qasm_simulator')
t_qc = transpile(bv_circuit, simulator)
qobj = assemble(t_qc)
result = simulator.run(qobj).result()

# Get the result counts and plot the histogram
counts = result.get_counts()
plot_histogram(counts)
```

Listing 13: Qiskit code for Bernstein-Vazirani Problem generated by GPT4o in the zero-shot setting.

This code, although adopts an older version of Qiskit syntax, can be compiled and run successfully with the correct answer. At first glance, this indicates that GPT4o has fully grasped the mechanism of the Bernstein-Vazirani Algorithm and the syntax of Qiskit. Similar performances can be observed with OpenQASM codes:

```
// OpenQASM 2.0
OPENQASM 2.0;
include "qelib1.inc";

// Declare 5 qubits for input and 1 ancilla qubit
qreg q[6];
creg c[5]; // classical register for measuring the result

// Step 1: Apply Hadamard gates to the input qubits
h q[0];
```

```
h q[1];
h q[2];
h q[3];
h q[4];

// Apply Hadamard to the ancillary qubit and set it to |1>
x q[5];
h q[5];

// Step 2: Oracle implementation (CNOT gates based on secret string s =
    ↪ 10110)
cx q[0], q[5]; // Secret string bit 1
// No cx for q[1] because the secret string bit is 0
cx q[2], q[5]; // Secret string bit 1
cx q[3], q[5]; // Secret string bit 1
// No cx for q[4] because the secret string bit is 0

// Step 3: Apply Hadamard gates to the input qubits again
h q[0];
h q[1];
h q[2];
h q[3];
h q[4];

// Step 4: Measure the qubits to get the result
measure q[0] -> c[0];
measure q[1] -> c[1];
measure q[2] -> c[2];
measure q[3] -> c[3];
measure q[4] -> c[4];
```

Listing 14: OpenQASM code for Bernstein-Vazirani Problem generated by GPT4o.

However, if we ask the GPT4o model to implement the code for a given secret string, for example $s = 000101$, the model fails to apply the CX gate on the correct qubits.

```
OPENQASM 3.0;
include "stdgates.inc";

gate Oracle _gate_q_0, _gate_q_1, _gate_q_2, _gate_q_3, _gate_q_4,
    ↪ _gate_q_5, _gate_q_6 {
  // Apply CX gates according to the secret string "000101"
  cx _gate_q_2, _gate_q_6;
  cx _gate_q_5, _gate_q_6;
}

bit[6] c;
qubit[7] q;

// Apply Hadamard gates to the first 6 qubits
h q[0];
h q[1];
h q[2];
h q[3];
h q[4];
h q[5];

// Prepare the ancilla qubit in the |1> state and apply a Hadamard gate
x q[6];
h q[6];

// Call the oracle
Oracle q[0], q[1], q[2], q[3], q[4], q[5], q[6];

// Apply Hadamard gates to the first 6 qubits again
```

```
h q[0];
h q[1];
h q[2];
h q[3];
h q[4];
h q[5];

// Measure the first 6 qubits
c[0] = measure q[0];
c[1] = measure q[1];
c[2] = measure q[2];
c[3] = measure q[3];
c[4] = measure q[4];
c[5] = measure q[5];
```

Listing 15: OpenQASM code for Bernstein-Vazirani Problem with s = 000101 in few-shot setting.

We reminded the model of the mechanism with the following prompt:

**Reminder Prompt**

Remember that in the construction of oracles for Bernstein-Vazirani Problem, you should apply cx gates where the control qubits are where the string bit equals 1, and the target qubit is the ancillary qubit.

There was no improvement. We further asked the model about the indices of the bits in the string which equal 1. It provided the following answer:

**GPT4o's answer for counting indices**

In the secret string s="000101", the bits that equal 1 are found at the following indices (0-based):
Index 2
Index 5
These indices correspond to the positions where the string has a '1'.

This reveals the incapability for GPT models to count the bit of 1 and apply corresponding gates to construct the oracle. Considering that Qiskit has implemented tutorial codes for several representative quantum algorithms, the seemingly outstanding performance of quantum algorithm code generation might be a deceptive result caused by data contamination. Therefore, only a carefully designed quantum algorithm dataset can avoid the effects of data contamination, allowing for an effective evaluation of the model's genuine capability in quantum algorithm design and implementation. This dataset is also meaningful for testing general AI code generation and syntax learning, where no existing AI dataset could substitute us.

