# OpenReview forum: "QCircuitNet: A Large-Scale Hierarchical Dataset for Quantum Algorithm Design"
_ICLR.cc/2025/Conference — Submitted to ICLR 2025_

### Official Review · Reviewer_QgHZ · 2024-11-02

**Soundness:** 2
**Presentation:** 2
**Contribution:** 2
**Rating:** 3
**Confidence:** 5

**Summary:**

This paper presents QCircuitNet, the benchmark and test dataset for evaluating AI’s ability to design and implement quantum algorithms in quantum circuit codes.

**Strengths:**

+ Interesting direction of using AI/LLM for developing quantum circuits for a particular quantum algorithm
+ Dataset can be useful to some extent

**Weaknesses:**

+ Not clear what exactly are the new contributions in addition to the “run_and_analyze” function.

**Questions:**

This paper claims that existing application benchmarks fail as a dataset for AI because they did not capture the design patterns of each algorithm, ignore post-processing and construction of different oracles. However, after reading the paper, I could not find what exactly are the “design patterns” of a quantum algorithm. The paper did not clarify any circuit features, or algorithm features, or device features.

Meanwhile, It seems this work only focuses on oracle-based quantum algorithms, which represents a very small group of quantum algorithms that were well-known and developed many years ago (and is also not the main focus of the quantum computing/algorithm community).

It seems to me this claimed ‘dataset contribution’ is only a different way of selling the application benchmarks by (i) collecting the standard descriptions of particular quantum algorithms from text or existing paper; (ii) running in Qiskit-Aer to obtain standard output as the reference; (iii) adding metric measurement functions for the feedback/reward/gradient, which can be easily achieved using the MQTBench templates online. With that, I don’t think the technical contribution is sufficient for an ICLR publication.

Last but not least, the word ‘post-processing’ is confusing, as in quantum computing, post-processing usually refers to error-mitigation.

---

> ### Author Response · Authors · 2024-11-17
>
> **1. This paper claims that existing application benchmarks fail as a dataset for AI because they did not capture the design patterns of each algorithm, ignore post-processing and construction of different oracles. However, after reading the paper, I could not find what exactly are the “design patterns” of a quantum algorithm. The paper did not clarify any circuit features, or algorithm features, or device features. It seems to me this claimed ‘dataset contribution’ is only a different way of selling the application benchmarks by (i) collecting the standard descriptions of particular quantum algorithms from text or existing paper; (ii) running in Qiskit-Aer to obtain standard output as the reference; (iii) adding metric measurement functions for the feedback/reward/gradient, which can be easily achieved using the MQTBench templates online. With that, I don’t think the technical contribution is sufficient for an ICLR publication.**
>
> We respectfully disagree with the reviewer and clarify our contribution with Simon's Problem as a concrete example. Imagine that you have never learned this algorithm before, what do you have in hand? A description to the problem, which states the problem setting; a black-box oracle, which you can use but without knowledge of its circuit implementation. How do we know if you have successfully designed the algorithm? As long as you can correctly apply the Hadamard gates before and after the oracle to construct a quantum circuit, and write the classical procedures of solving linear equations to obtain the answer, you have finished the design of this quantum algorithm. This process emphasizes on two points: 1. **You must not know the circuit implementation for the oracle.** Otherwise, you can directly deduce the secret string from the oracle structure. 2. To derive answer to the original problem, you have to provide **a classical procedure** which interprets the measurement results **in addition to the quantum circuit**. However, none of the existing benchmarks achieve these two points.
>
> (1) Existing benchmarks do not separate the implementation of the oracle from the algorithm circuit (Hadamard gates in Simon's case). They typically look like this:
>
> ```
> OPENQASM 3.0;
> include "stdgates.inc";
> bit[2] c;
> qubit[4] q;
> h q[0];
> h q[1];
> cx q[0], q[2];
> cx q[1], q[3];
> cx q[0], q[2];
> cx q[0], q[3];
> x q[2];
> x q[3];
> h q[0];
> h q[1];
> c[0] = measure q[0];
> c[1] = measure q[1];
> ```
>
> However, what we want the model to generate looks like this:
>
> ```
> OPENQASM 3.0;
> include "stdgates.inc";
> bit[2] c;
> qubit[4] q;
> h q[0];
> h q[1];
> Oracle q[0], q[1], q[2], q[3];
> h q[0];
> h q[1];
> c[0] = measure q[0];
> c[1] = measure q[1];
> ```
> This brings another problem to verification. For all experimental platforms, a quantum circuit needs to be explicitly constructed to compile and run successfully, i.e., the oracle should be provided with exact gate implementation. To make the second code valid in OpenQASM grammar while keep the definition for Oracle in the black box, we use a separate "oracle.inc" file defining the "Oracle" gate and add the include "oracle.inc"; line, which guarantees the correctness of OpenQASM’s grammar while separating oracle from the algorithm circuit.
>
> (2) Existing benchmarks typically only provide quantum circuits, and leave out the classical procedure to interpret circuit measurement results. To bridge this gap, we write a separate module named "post-processing" function, for example the code to solve linear equations to derive the answer for Simon's Problem.

---

> ### Author Response · Authors · 2024-11-17
>
> (3) Fine-grained quantum circuits on the gate level result in extremely long code which is hard for LLMs to generate. For example multi-controlled X gate contains 45060 lines for qubit number n = 14 in OpenQASM format. On the one hand, to avoid decomposition of circuits into basis gates, we provide composite gates in the additional "customgates.inc". On the other hand, we rewrite the qasm codes with hierarchical definition for subroutines. For instance, in the phase estimation task with qubit number $n=6$, the direct output from Qiskit would be:
> ```
> OPENQASM 3.0;
> include "stdgates.inc";
> gate Psi _gate_q_0 {
>   x _gate_q_0;
> }
> gate CU_5010903776(_gate_p_0) _gate_q_0, _gate_q_1 {
>   p(1.9694939281977046) _gate_q_0;
>   cx _gate_q_0, _gate_q_1;
>   p(-1.9694939281977046) _gate_q_1;
>   cx _gate_q_0, _gate_q_1;
>   p(1.9694939281977046) _gate_q_1;
> }
> gate IQFT _gate_q_0, _gate_q_1, _gate_q_2, _gate_q_3, _gate_q_4, _gate_q_5 {
>   swap _gate_q_0, _gate_q_5;
>   swap _gate_q_1, _gate_q_4;
>   swap _gate_q_2, _gate_q_3;
>   h _gate_q_0;
>   cp(-pi/2) _gate_q_0, _gate_q_1;
>   h _gate_q_1;
>   cp(-pi/4) _gate_q_0, _gate_q_2;
>   cp(-pi/2) _gate_q_1, _gate_q_2;
>   h _gate_q_2;
>   cp(-pi/8) _gate_q_0, _gate_q_3;
>   cp(-pi/4) _gate_q_1, _gate_q_3;
>   cp(-pi/2) _gate_q_2, _gate_q_3;
>   h _gate_q_3;
>   cp(-pi/16) _gate_q_0, _gate_q_4;
>   cp(-pi/8) _gate_q_1, _gate_q_4;
>   cp(-pi/4) _gate_q_2, _gate_q_4;
>   cp(-pi/2) _gate_q_3, _gate_q_4;
>   h _gate_q_4;
>   cp(-pi/32) _gate_q_0, _gate_q_5;
>   cp(-pi/16) _gate_q_1, _gate_q_5;
>   cp(-pi/8) _gate_q_2, _gate_q_5;
>   cp(-pi/4) _gate_q_3, _gate_q_5;
>   cp(-pi/2) _gate_q_4, _gate_q_5;
>   h _gate_q_5;
> }
> bit[6] c;
> qubit[7] q;
> Psi q[6];
> h q[0];
> h q[1];
> h q[2];
> h q[3];
> h q[4];
> h q[5];
> CU_5010903776(3.9389878563954093) q[0], q[6];
> CU_5010903776(3.9389878563954093) q[1], q[6];
> CU_5010903776(3.9389878563954093) q[1], q[6];
> CU_5010903776(3.9389878563954093) q[2], q[6];
> CU_5010903776(3.9389878563954093) q[2], q[6];
> CU_5010903776(3.9389878563954093) q[2], q[6];
> CU_5010903776(3.9389878563954093) q[2], q[6];
> CU_5010903776(3.9389878563954093) q[3], q[6];
> CU_5010903776(3.9389878563954093) q[3], q[6];
> CU_5010903776(3.9389878563954093) q[3], q[6];
> CU_5010903776(3.9389878563954093) q[3], q[6];
> CU_5010903776(3.9389878563954093) q[3], q[6];
> CU_5010903776(3.9389878563954093) q[3], q[6];
> CU_5010903776(3.9389878563954093) q[3], q[6];
> CU_5010903776(3.9389878563954093) q[3], q[6];
> CU_5010903776(3.9389878563954093) q[4], q[6];
> CU_5010903776(3.9389878563954093) q[4], q[6];
> CU_5010903776(3.9389878563954093) q[4], q[6];
> CU_5010903776(3.9389878563954093) q[4], q[6];
> CU_5010903776(3.9389878563954093) q[4], q[6];
> CU_5010903776(3.9389878563954093) q[4], q[6];
> CU_5010903776(3.9389878563954093) q[4], q[6];
> CU_5010903776(3.9389878563954093) q[4], q[6];
> CU_5010903776(3.9389878563954093) q[4], q[6];
> CU_5010903776(3.9389878563954093) q[4], q[6];
> CU_5010903776(3.9389878563954093) q[4], q[6];
> CU_5010903776(3.9389878563954093) q[4], q[6];
> CU_5010903776(3.9389878563954093) q[4], q[6];
> CU_5010903776(3.9389878563954093) q[4], q[6];
> CU_5010903776(3.9389878563954093) q[4], q[6];
> CU_5010903776(3.9389878563954093) q[4], q[6];
> CU_5010903776(3.9389878563954093) q[5], q[6];
> CU_5010903776(3.9389878563954093) q[5], q[6];
> CU_5010903776(3.9389878563954093) q[5], q[6];
> CU_5010903776(3.9389878563954093) q[5], q[6];
> CU_5010903776(3.9389878563954093) q[5], q[6];
> CU_5010903776(3.9389878563954093) q[5], q[6];
> CU_5010903776(3.9389878563954093) q[5], q[6];
> CU_5010903776(3.9389878563954093) q[5], q[6];
> CU_5010903776(3.9389878563954093) q[5], q[6];
> CU_5010903776(3.9389878563954093) q[5], q[6];
> CU_5010903776(3.9389878563954093) q[5], q[6];
> CU_5010903776(3.9389878563954093) q[5], q[6];
> CU_5010903776(3.9389878563954093) q[5], q[6];
> CU_5010903776(3.9389878563954093) q[5], q[6];
> CU_5010903776(3.9389878563954093) q[5], q[6];
> CU_5010903776(3.9389878563954093) q[5], q[6];
> CU_5010903776(3.9389878563954093) q[5], q[6];
> CU_5010903776(3.9389878563954093) q[5], q[6];
> CU_5010903776(3.9389878563954093) q[5], q[6];
> CU_5010903776(3.9389878563954093) q[5], q[6];
> CU_5010903776(3.9389878563954093) q[5], q[6];
> CU_5010903776(3.9389878563954093) q[5], q[6];
> CU_5010903776(3.9389878563954093) q[5], q[6];
> CU_5010903776(3.9389878563954093) q[5], q[6];
> CU_5010903776(3.9389878563954093) q[5], q[6];
> CU_5010903776(3.9389878563954093) q[5], q[6];
> CU_5010903776(3.9389878563954093) q[5], q[6];
> CU_5010903776(3.9389878563954093) q[5], q[6];
> CU_5010903776(3.9389878563954093) q[5], q[6];
> CU_5010903776(3.9389878563954093) q[5], q[6];
> CU_5010903776(3.9389878563954093) q[5], q[6];
> CU_5010903776(3.9389878563954093) q[5], q[6];
> IQFT q[0], q[1], q[2], q[3], q[4], q[5];
> c[0] = measure q[0];
> c[1] = measure q[1];
> c[2] = measure q[2];
> c[3] = measure q[3];
> c[4] = measure q[4];
> c[5] = measure q[5];
> ```
> This raw output is impossible for LLMs to generate accurately.

---

> ### Author Response · Authors · 2024-11-17
>
> After our restructure process, which renamed the CU gate and created hierarchical definition for the exponential gates, it became:
> ```
> OPENQASM 3.0;
> include "stdgates.inc";
> include "oracle.inc";
> gate CU_1 _gate_q_0, _gate_q_1 {
>   pow(2) @ CU_0 _gate_q_0, _gate_q_1;
> }
> gate CU_2 _gate_q_0, _gate_q_1 {
>   pow(2) @ CU_1 _gate_q_0, _gate_q_1;
> }
> gate CU_3 _gate_q_0, _gate_q_1 {
>   pow(2) @ CU_2 _gate_q_0, _gate_q_1;
> }
> gate CU_4 _gate_q_0, _gate_q_1 {
>   pow(2) @ CU_3 _gate_q_0, _gate_q_1;
> }
> gate CU_5 _gate_q_0, _gate_q_1 {
>   pow(2) @ CU_4 _gate_q_0, _gate_q_1;
> }
> gate IQFT _gate_q_0, _gate_q_1, _gate_q_2, _gate_q_3, _gate_q_4, _gate_q_5 {
>   swap _gate_q_0, _gate_q_5;
>   swap _gate_q_1, _gate_q_4;
>   swap _gate_q_2, _gate_q_3;
>   h _gate_q_0;
>   cp(-pi/2) _gate_q_0, _gate_q_1;
>   h _gate_q_1;
>   cp(-pi/4) _gate_q_0, _gate_q_2;
>   cp(-pi/2) _gate_q_1, _gate_q_2;
>   h _gate_q_2;
>   cp(-pi/8) _gate_q_0, _gate_q_3;
>   cp(-pi/4) _gate_q_1, _gate_q_3;
>   cp(-pi/2) _gate_q_2, _gate_q_3;
>   h _gate_q_3;
>   cp(-pi/16) _gate_q_0, _gate_q_4;
>   cp(-pi/8) _gate_q_1, _gate_q_4;
>   cp(-pi/4) _gate_q_2, _gate_q_4;
>   cp(-pi/2) _gate_q_3, _gate_q_4;
>   h _gate_q_4;
>   cp(-pi/32) _gate_q_0, _gate_q_5;
>   cp(-pi/16) _gate_q_1, _gate_q_5;
>   cp(-pi/8) _gate_q_2, _gate_q_5;
>   cp(-pi/4) _gate_q_3, _gate_q_5;
>   cp(-pi/2) _gate_q_4, _gate_q_5;
>   h _gate_q_5;
> }
> bit[6] c;
> qubit[7] q;
> Psi q[6];
> h q[0];
> h q[1];
> h q[2];
> h q[3];
> h q[4];
> h q[5];
> CU_0 q[0], q[6];
> CU_1 q[1], q[6];
> CU_2 q[2], q[6];
> CU_3 q[3], q[6];
> CU_4 q[4], q[6];
> CU_5 q[5], q[6];
> IQFT q[0], q[1], q[2], q[3], q[4], q[5];
> c[0] = measure q[0];
> c[1] = measure q[1];
> c[2] = measure q[2];
> c[3] = measure q[3];
> c[4] = measure q[4];
> c[5] = measure q[5];
> ```
> Which significantly reduces the difficulty of generation.
>
> (4) The verification function needs to be carefully designed to accommodate these changes. Since Qiskit does not support importation of non-standard gate libraries such as "oracle.inc" and "customgates.inc" currently, we explicitly integrates the oracle / gate definition library with output algorithm circuit in verification function. Moreover, the function is not restricted to checking the **measurement result** of the quantum circuit, but also the correctness of the classical post-processing function. It is designed to be generic so that novel implementations differ from the reference would also be labeled as correct.
>
> There are many other efforts we have made to tailor the algorithm design task for LLMs, and the engineering efforts are highly non-trivial to be compatible across Qiskit, OpenQASM, and LLM platforms. We recognize MQTBench along with other benchmarks as important work for evaluating the quantum software tools on the implementation side, but they are infeasible to formulate the task of quantum algorithm design for LLMs from the theoretical computer science perspective. As recognized by other reviewers, QCircuitNet provides a structured framework which encompasses the key features of quantum algorithm designs, with good care of how to properly encode them. We believe it is a meaningful first step for the important problem of AI-aided quantum algorithm design.
>
> **2. Meanwhile, It seems this work only focuses on oracle-based quantum algorithms, which represents a very small group of quantum algorithms that were well-known and developed many years ago (and is also not the main focus of the quantum computing/algorithm community).**
>
> We respectfully disagree with the reviewer. On the one hand, oracle-based quantum algorithms are the origin and still an active research mainstream of theoretical quantum computing. On the other hand, we have included quantum information and random circuit synthesis as major categories of our dataset, which goes beyond oracle-based quantum algorithms.
>
> **3. Last but not least, the word ‘post-processing’ is confusing, as in quantum computing, post-processing usually refers to error-mitigation.**
>
> We thank the reviewer for the feedback. We initially chose this word to emphasize that a quantum algorithm constitutes not only the quantum circuit, but also the interpretation of measurement results to derive the answer to the original problem. We will consider renaming this component to avoid confusion.

---

### Official Review · Reviewer_o2k6 · 2024-11-03

**Soundness:** 2
**Presentation:** 3
**Contribution:** 2
**Rating:** 5
**Confidence:** 2

**Summary:**

This paper introduces QCircuitNet, a benchmark dataset specifically designed to evaluate AI's capability in designing and implementing quantum algorithms as quantum circuit codes.

**Strengths:**

1. A proposed method effectively captures quantum algorithms, situated between pure mathematical formulas and natural language.

**Weaknesses:**

1. The dataset contains only classic algorithms and lacks generalizability.
2. Lacks tests for code completion.

**Questions:**

1. Why call "QCircuitNet"? I mean it is more like a neural network rather than a dataset or benchmark. This name is confusing.
2. How many algorithms are currently included in the dataset? Does each algorithm contain test data with different number of qbits?
3. "The total computation cost is approximately equivalent to two days on an A100 GPU." you mean to test one model or all the experiments add up cost 2days?
4. Can you incorporate "chain of thought" in this benchmark?

---

> ### Author Response · Authors · 2024-11-17
>
> We thank the reviewer for the feedback and further clarify the points raised:
>
> **1. The dataset contains only classic algorithms and lacks generalizability.**
>
> In QCircuitNet, in addition to representative quantum algorithms, we have implemented advanced algorithms such as the Generalized Simon's Problem, which have been actively studied in recent years. This inclusion effectively showcases the generalizability of our framework to a broader spectrum of quantum algorithms beyond classic cases. More details can be found in Section 4.1.2 of the original
> manuscript.
>
> **2. Lacks tests for code completion.**
>
> We would like to clarify that the verification function explicitly checks if the code is complete and executable, and returns -1 if the code is not complete or contains grammar errors.
>
> **3. Why call "QCircuitNet"? I mean it is more like a neural network rather than a dataset or benchmark. This name is confusing.**
>
> The name QCircuitNet was originally inspired by ImageNet, a foundational dataset for image classification which significantly contributed to the advancement of deep learning. We thank the reviewer for the advice and will consider renaming the dataset to avoid confusion.
>
> **4. How many algorithms are currently included in the dataset? Does each algorithm contain test data with different number of qbits?**
>
> There are 26 different quantum algorithm design / oracle construction / circuit synthesis tasks. Yes, each task contains test data with different qubit numbers. Moreover, for quantum algorithm design task, each algorithm contains test data with different test oracles. For oracle construction, each task includes different logical functions to encode, such as different secret strings in Simon's Problem. For random circuit synthesis task, each category includes a large number of random circuits using the provided gate set.
>
> **5. "The total computation cost is approximately equivalent to two days on an A100 GPU." you mean to test one model or all the experiments add up cost 2days?**
>
> We apologize for the confusion caused. The computation cost of approximately two days on an A100 GPU refers to the total time required for all the experiments conducted on the GPU. This estimate does not include the time spent accessing GPT-4 and GPT-3.5 via the OpenAI API, nor does it include the additional time required for verification problem computations, which were handled on the CPU.
>
> **6. Can you incorporate "chain of thought" in this benchmark?**
>
> We thank the reviewer for this valuable question. In terms of code generation [1, 2, 3], annotations might be more common than chain-of-thought prompting. For quantum algorithm design, it seems challenging to create meaningful intermediate steps for the reference due to the *lack of compositionality*. Thus, aside from in-context Bayesian optimization, chain-of-thought has limited capacity to serve as a scratchpad or template, which may reduce its potential benefits. Nevertheless, we thank the reviewer for the suggestion and regard this as an interesting direction worth exploring through more systematic investigation in future work.
>
> [1] Roziere, Baptiste, Jonas Gehring, Fabian Gloeckle, Sten Sootla, Itai Gat, Xiaoqing Ellen Tan, Yossi Adi et al. "Code llama: Open foundation models for code." arXiv preprint arXiv:2308.12950 (2023).
>
> [2] Li, Raymond, Loubna Ben Allal, Yangtian Zi, Niklas Muennighoff, Denis Kocetkov, Chenghao Mou, Marc Marone et al. "Starcoder: may the source be with you!." arXiv preprint arXiv:2305.06161 (2023).
>
> [3] Guo, Daya, Qihao Zhu, Dejian Yang, Zhenda Xie, Kai Dong, Wentao Zhang, Guanting Chen et al. "DeepSeek-Coder: When the Large Language Model Meets Programming--The Rise of Code Intelligence." arXiv preprint arXiv:2401.14196 (2024).

---

> > ### Comment · Reviewer_o2k6 · 2024-12-02
> >
> > Thank you for your responses. Some questions have been answered. However, I still have concerns about the broad applicability of the algorithms involved in this dataset, and I don't fully understand the necessity of this benchmark. My confidence score somehow reflect this point.

---

### Official Review · Reviewer_z41s · 2024-11-04

**Soundness:** 4
**Presentation:** 3
**Contribution:** 2
**Rating:** 6
**Confidence:** 4

**Summary:**

The authors present a new dataset, QCircuitNet, for Quantum Algorithm Design, aimed at enhancing the design and implementation of quantum algorithms using LLMs. The authors a structured framework that allow the LLMs to apply to the quantum algorithm discovery. QCircuitNet features built-in functions for automatic validation and verification of algorithms, supporting iterative evaluation without human intervention. Experiments results highlight the evaluation of LLMs for quantum algorithm discovery with dataset, showcasing its potential as a valuable resource in the field.

**Strengths:**

1. QCircuitNet is the first dataset specifically designed for evaluating LLMs in quantum algorithm design.

2. The authors provide a structured framework that encapsulates key features of quantum algorithm design, allowing LLMs to work effectively with complex quantum tasks.

3. The paper presents valuable experimental findings regarding the performance of LLMs in quantum algorithm discovery.

**Weaknesses:**

1. The paper primarily addresses quantum circuit design as a language modeling task, which may limit the scope of quantum algorithm generation and overlook other important methodologies.

2. The authors do not address Variational Quantum Algorithms, which are significant in near term quantum computing, indicating a gap in the dataset's comprehensiveness.

3. Given the limited research on LLMs for quantum circuits, the necessity of creating this dataset at this stage may be questioned.

4. The evaluation metrics used are heavily influenced by natural language processing. It raises the question of whether there might be a more intrinsic approach to integrating quantum into these metrics.

**Questions:**

1. Why did you choose to focus primarily on quantum circuit design as a language modelling task?

2. Are there alternative methodologies or frameworks that could also be explored for quantum algorithm discovery with your dataset?

2. What is your rationale for excluding VQAs from QCircuitNet? How do you see their importance in the context of quantum algorithm design?

3. Have you considered developing metrics that are more aligned with quantum computing?

---

> ### Author Response · Authors · 2024-11-17
>
> **1. The paper primarily addresses quantum circuit design as a language modeling task, which may limit the scope of quantum algorithm generation and overlook other important methodologies. Why did you choose to focus primarily on quantum circuit design as a language modeling task? Are there alternative methodologies or frameworks that could also be explored for quantum algorithm discovery with your dataset?**
>
> We chose to approach quantum algorithm design from a programming language perspective, framing it similarly to the classical *code generation* task. Alternative methodologies, such as using natural language descriptions, could result in verbosity and ambiguity, making it challenging to represent algorithms clearly. While mathematical formulas provide precision and conciseness, they are difficult to verify automatically. By modeling quantum algorithm design as code generation, we achieve a precise representation of quantum algorithms, facilitate an automatic verification process, and bridge the gap between theoretical design and concrete circuit implementations.
>
> **2. The authors do not address Variational Quantum Algorithms, which are significant in near term quantum computing, indicating a gap in the dataset's comprehensiveness. What is your rationale for excluding VQAs from QCircuitNet? How do you see their importance in the context of quantum algorithm design?**
>
> In this dataset, we focus primarily on universal quantum algorithms, a foundational category in quantum computing. While we acknowledge the importance of variational quantum algorithms, particularly for near-term applications, they were not initially included due to ambiguity in verifying their correctness in a definitive right-or-wrong manner. Variational algorithms are typically assessed through performance metrics, such as the closeness of the produced state and eigenvalue to the ground state and ground energy in the variational quantum eigensolver. We thank the reviewer for the suggestion and are open to including variational quantum algorithms in future work by incorporating performance-based evaluation metrics as a verification function.
>
> **3. Given the limited research on LLMs for quantum circuits, the necessity of creating this dataset at this stage may be questioned.**
>
> Thank you for raising this insightful point. We acknowledge that research on LLMs for quantum circuits is still in its early stages. However, this is precisely why we believe the creation of a specialized dataset is both timely and essential. History has shown that seminal datasets play a crucial role in sparking breakthroughs in emerging fields. A well-known example is ImageNet, which was introduced during the early days of computer vision, provided a standardized benchmark, and ultimately fueled advancements in deep learning and transformed the field. Similarly, our dataset is intended to serve as a first step towards the important problem of AI-aided quantum algorithm design, hopefully facilitating research in this direction.
>
>
> **4. The evaluation metrics used are heavily influenced by natural language processing. It raises the question of whether there might be a more intrinsic approach to integrating quantum into these metrics. Have you considered developing metrics that are more aligned with quantum computing?**
>
> We appreciate the reviewer’s feedback. We have indeed incorporated metrics aligned with quantum computing; specifically, we designed the verification score as a key evaluation metric. For example, in state preparation tasks such as generating the W state, the quantum state fidelity is calculated as the verification score. We believe that combining NLP metrics, such as BLEU, with quantum-specific metrics like verification score provides a comprehensive evaluation framework.

---

> > ### Comment · Reviewer_z41s · 2024-11-22
> >
> > Framing of quantum algorithm design as akin to classical code generation may introduce some limitations, particularly in terms of dataset compatibility with other approaches. For instance, the classical code generation approach typically assumes discrete structures, while quantum algorithms often involve continuous variables, such as rotational gates in quantum Fourier transform. These continuous parameters are critical for capturing the full expressiveness of quantum algorithms, and a strict code generation framework may struggle to represent them intuitively. My concerns regarding Variational Quantum Algorithm problems align with this issue, as VQAs frequently rely on continuous parameter optimization.
> >
> > Thank you for your thoughtful reply. However, I remain unconvinced and strongly doubt that future research on Quantum Algorithm Design will predominantly focus on LLMs. While the dataset may be well-suited for LLMs, it may not cater effectively to other approaches, potentially limiting broader advancements in this area.

---

> > > ### Author Response · Authors · 2024-11-29
> > >
> > > We greatly appreciate the reviewer for the follow-up feedback and hope the following response may address the concerns.
> > >
> > > **1. Compatibility with VQA:** To demonstrate that QCircuitNet is compatible with Variational Quantum Algorithms, we have carried out additional experiments with two concrete examples: implementing the Variational Quantum Eigensolver (VQE) to find the ground-state energy of a given Hamiltonian and the Quantum Approximate Optimization Algorithm (QAOA) to find the maximum cut of a given graph. We ask the LLM to design the VQE ansatz in QASM and implement the `run_and_analyze` function in Python, which optimizes the circuit parameters and computes the final results.
> > > ### **VQE task:**
> > > To evaluate correctness, we compare the energy obtained from the LLM-designed ansatz with the ground truth and calculate a score:
> > > $$
> > > 1 - \frac{|LLM\ result\ energy - expected\ energy|}{|expected\ energy|}
> > > $$
> > > The verification is divided into the following cases:
> > > - (1) **QASM Syntax Error**: If the model-generated QASM has syntax errors, the score is −1.
> > > - (2) **Python Syntax Error**: If the QASM is valid but the Python code for `run_and_analyze` has syntax errors, the QASM output is evaluated using a ground truth implementation of `run_and_analyze`.
> > >     - If the result matches the ground truth, the score is $0.5\times Score$ (half the full score).
> > >     - If the result is incorrect, the score is 0.
> > >     - If the evaluation encounters further syntax errors, the score is −1.
> > > - (3) **Correct QASM and Python**: If both the QASM and Python code are correct and produce accurate results, the model receives the full score.
> > >
> > > ### **QAOA task:**
> > > The verification process is divided into the following cases:
> > > - (1) **QASM Syntax Error**: If the model-generated QASM has syntax errors, the score is −1.
> > > - (2) **Python Syntax Error**: If the QASM is valid but the Python code for `run_and_analyze` has syntax errors, the QASM output is evaluated using a ground truth implementation of `run_and_analyze`.
> > >     - If the result matches the ground truth partition, the score is $0.5$.
> > >     - If the result is incorrect, the score is 0.
> > >     - If the evaluation encounters further syntax errors, the score is −1.
> > > - (3) **Correct QASM and Python**:
> > > 	- If the partition matches the ground truth, the score is 1.
> > > 	- If the partition is incorrect, the score is 0.25.
> > >
> > > The results are as follows:
> > >
> > > Table 1: Standard Error of BLEU scores in variational circuit algorithm design
> > >
> > > | Model                      | Shot | VQE               | QAOA              | Average            |
> > > |----------------------------|------|-------------------|-------------------|--------------------|
> > > | gpt-4o-2024-05-13          | 1    | 12.7935(±2.8579)   | 18.1658(±1.0567)  | 15.4797 |
> > > | gpt-4o-2024-05-13          | 3    | 11.7136(±2.6834)  | 18.1072(±1.2072)  | 14.9104  |
> > > | Meta-Llama-3-8B            | 1    | 14.6278(±1.0701)  | 4.3151(±0.4370)   | 9.4714  |
> > > | Meta-Llama-3-8B            | 3    | 16.0207(±1.7138)  | 5.9892(±0.9450)   | 11.0050  |
> > > | gpt-3.5-turbo-0125         | 1    | 11.0529(±4.6120)  | 8.1221(±0.8570)   | 9.5875  |
> > > | gpt-3.5-turbo-0125         | 3    | 23.6283(±8.4819)  | 10.8345(±1.0061)  | 17.0261  |
> > > | Phi-3-medium-128k-instruct | 1    | 21.7502(±6.0640)  | 12.3021(±1.4899)  | 17.0261 |
> > > | Phi-3-medium-128k-instruct | 3    | 17.7635(±4.3053)  | 14.0514(±1.7597)  | 15.9074 |
> > > | Mistral-7B-v0.3            | 1    | 15.5163(±8.2261)  | 17.0759(±1.9467)  | 16.2961  |
> > > | Mistral-7B-v0.3            | 3    | 32.3204(±1.9443)  | 9.6164(±1.4423)   | 20.9684 |
> > >
> > >
> > > Table 2: Standard Error of verification function scores in variational circuit algorithm design
> > >
> > > | Model                      | Shot | VQE                  | QAOA                 | Average            |
> > > |----------------------------|------|----------------------|----------------------|--------------------|
> > > | gpt-4o-2024-05-13          | 1    | 0.2874(±0.0655)      | 0.1667(±0.2357)     | 0.2270  |
> > > | gpt-4o-2024-05-13          | 3    | 0.2270(±0.0209)      | 0.0556(±0.2693)     | 0.1413  |
> > > | Meta-Llama-3-8B            | 1    | -1.0000(±0.0000)     | -1.0000(±0.0000)     | -1.0000  |
> > > | Meta-Llama-3-8B            | 3    | -1.0000(±0.0000)     | -1.0000(±0.0000)     | -1.0000  |
> > > | gpt-3.5-turbo-0125         | 1    | -0.7300(±0.2700)     | -1.0000(±0.0000)     | -0.8650  |
> > > | gpt-3.5-turbo-0125         | 3    | -1.0000(±0.0000)     | -1.0000(±0.0000)     | -1.0000  |
> > > | Phi-3-medium-128k-instruct | 1    | -1.0000(±0.0000)     | -1.0000(±0.0000)     | -1.0000  |
> > > | Phi-3-medium-128k-instruct | 3    | -1.0000(±0.0000)     | -1.0000(±0.0000)     | -1.0000  |
> > > | Mistral-7B-v0.3            | 1    | -1.0000(±0.0000)     | -1.0000(±0.0000)     | -1.0000  |
> > > | Mistral-7B-v0.3            | 3    | -0.8125(±0.1875)     | -1.0000(±0.0000)     | -0.9063  |

---

> > > ### Author Response · Authors · 2024-11-29
> > >
> > > **2. Analogy to classical code generation:** On the one hand, quantum algorithms bear the discrete structure from the view of sequences of quantum gates. On the other hand, classical code generation can also be parametrized with continuous variables. Instead of strictly following classical code generation, we believe it is useful to borrow the **core idea of access to verification and unit tests**. We acknowledge that pure classical code generation techniques are insufficient for quantum algorithm design, which is exactly why we develop this benchmark to stimulate quantum-intrinsic methods.
> > >
> > > **3. Limitation of restrictions to LLM-based methods:** We initially chose LLMs due to their abundant pre-trained knowledge and strong capability for sequential modeling. Yet we are happy to explore the application of our dataset to other methodologies. Could the reviewer kindly specify what "other approaches" and "alternative methodologies or framework" refer to? We are happy to discuss whether QCircuitNet can be adapted to these settings.

---

### Official Review · Reviewer_uTSv · 2024-11-05

**Soundness:** 2
**Presentation:** 3
**Contribution:** 2
**Rating:** 5
**Confidence:** 3

**Summary:**

The paper introduces QCircuitNet, a novel dataset aimed at facilitating AI-driven quantum algorithm design. This dataset provides benchmarks and tools to evaluate the ability of LLMs to generate and validate quantum algorithms in quantum circuit code. QCircuitNet provides an automatic verification framework that ensures circuit validity.

**Strengths:**

The paper introduces QCircuitNet, and it provides a comprehensive structure, including benchmarks, automatic validation, and compatibility with various algorithms. Additionally, it offers a unique framework for large language models by formulating quantum algorithms as programming tasks.

**Weaknesses:**

See Questions.

**Questions:**

1. In QcircuitNet, what is the specific number of circuits, and what is the range of qubits? As a benchmark, the description of the dataset is unclear.

2. Directly using LLMs to design quantum algorithms remains challenging. For workflows like Figure 2, can LLMs only learn specific quantum algorithms? However, with the limited algorithms/circuit types in the current benchmark, it seems insufficient to evaluate LLMs’ ability to design arbitrary quantum algorithms.

3. The current setup relies on classical simulations for verification, which limits scalability and slows down processes, especially for higher qubit counts. If real quantum computers are used, significant noise will be present. In such cases, how can effective verification be ensured?

---

> ### Author Response · Authors · 2024-11-17
>
> We would like to thank the reviewer for recognizing that QCircuitNet provides a comprehensive structure and unique framework for LLMs by formulating quantum algorithms as programming tasks. We address the concerns as follows:
>
> **1. In QCircuitNet, what is the specific number of circuits, and what is the range of qubits? As a benchmark, the description of the dataset is unclear.**
>
> We thank the reviewer for the suggestion. The detailed information for circuit number and qubit range is provided as follows:
> | **Algorithm_Design**          | **Range** | **Circuit_Count** | **Algorithm_Design**  | **Range** | **Circuit_Count** |
> | ------------------------------ | --------- | ----------------- | ------------------------ | --------- | ----------------- |
> | bernstein_vazirani             | 2-30      | 511               | ghz_state                | 2-133     | 132               |
> | deutsch_jozsa                  | 2-30      | 569               | quantum_key_distribution | 20-50     | 620               |
> | generalized_simon_multi        | 2-30      | 671               | quantum_teleportation    | 3         | 100               |
> | generalized_simon_ternary      | 2-10      | 85                | random_number_generator  | 1-133     | 133               |
> | grover                         | 2-14      | 191               | superdense_coding        | 2         | 4                 |
> | phase_estimation               | 2-14      | 727               | swap_test                | 1-20      | 400               |
> | quantum_fourier_transformation | 2-30      | 511               | w_state                  | 2-133     | 132               |
> | simon                          | 2-30      | 671               |                          |           |                   |
>
> | **Oracle Construction**   | **Range** | **Circuit_Count** | **Random_Circuits** | **Range** | **Circuit_Count** |
> | ------------------------- | --------- | ----------------- | ------------------- | --------- | ----------------- |
> | grover_sudoku             | 9         | 1                 | clifford            | 2-12      | 38940             |
> | grover_triangle           | 3-14      | 60                | universal           | 2-12      | 51920             |
> | bernstein_vazirani        | 2-14      | 8191              |                     |           |                   |
> | deutsch_jozsa             | 2-14      | 8217              |                     |           |                   |
> | diffusion_operator        | 2-14      | 13                |                     |           |                   |
> | generalized_simon_multi   | 2-30      | 671               |                     |           |                   |
> | generalized_simon_ternary | 2-7       | 3364              |                     |           |                   |
> | grover                    | 2-14      | 8191              |                     |           |                   |
> | simon                     | 2-14      | 3541              |                     |           |                   |
>
>
> | **Category**         | **Circuit Count** |
> |----------------------|-------------------|
> | Total Circuits       | 128566            |
> | Algorithm Design     | 5457              |
> | Oracle Construction  | 32249             |
> | Random Circuits      | 90860             |

---

> ### Author Response · Authors · 2024-11-17
>
> **2. Directly using LLMs to design quantum algorithms remains challenging. For workflows like Figure 2, can LLMs only learn specific quantum algorithms? However, with the limited algorithms/circuit types in the current benchmark, it seems insufficient to evaluate LLMs’ ability to design arbitrary quantum algorithms.**
>
> We thank the reviewer for the valuable feedback. In this benchmark, we focus on universal quantum algorithms, covering most of the established and important quantum algorithms and primitives. The limited variety of algorithm types is an inherent characteristic of quantum computing rather than the incompleteness of the dataset. Our experiments demonstrate that the algorithms included effectively reveal the models' current capabilities and limitations. This is analogous to how the MATH dataset [1] assesses a model’s mathematical skills with high school competition problems, while evaluating a model's capacity for 'arbitrary' mathematical problems would also be challenging.
>
> The workflow in Figure 2 is not restricted to specific quantum algorithms. This few-shot learning framework demonstrates both the expected input-output format and provides concrete examples to guide algorithm design. We have structured this framework to be as general as possible, allowing it to accommodate new quantum algorithms whenever a suitable problem description, oracle file (.inc) if needed, quantum circuit code, post-processing, and verification function are available. If we focus solely on the verification score, the ground truth implementation of quantum circuit codes and post-processing functions can be omitted, enabling application to an even broader set of scenarios.
>
> **3. The current setup relies on classical simulations for verification, which limits scalability and slows down processes, especially for higher qubit counts. If real quantum computers are used, significant noise will be present. In such cases, how can effective verification be ensured?**
>
> We appreciate the reviewer's question. Please see general response to "scalability".
>
> [1] Hendrycks, Dan, Collin Burns, Saurav Kadavath, Akul Arora, Steven Basart, Eric Tang, Dawn Song, and Jacob Steinhardt. "Measuring mathematical problem solving with the math dataset." arXiv preprint arXiv:2103.03874 (2021).

---

> ### Comment · Reviewer_uTSv · 2024-11-29
>
> Thank you for your detailed reply. Some concerns have been addressed, but I still have concerns about the current practicalities of the work.

---

### Author Response · Authors · 2024-11-17
**General Response**

We would like to thank the reviewers for recognizing that QCircuitNet is the first dataset which provides a comprehensive framework for benchmarking quantum algorithm design with LLMs. The main concerns focus on the following aspects:
- **Generalizability to advanced quantum algorithms**: One core consideration to design the framework of QCircuitNet is to ensure easy extension to advanced quantum algorithms. In Section 4.1.2 of the original manuscript, we have provided a concrete example by implementing Generalized Simon's Problem, a more advanced version of the standard Simon’s problem and an active area of research in recent years [1, 2]. It effectively demonstrates the generalizability of QCircuitNet to advanced algorithms.
- **Scalability**: We choose classical simulation over real quantum computers as default verification methods also due to consideration of noises at NISQ era. As quantum hardware advances, these noise-related issues may gradually be alleviated. While we acknowledge that the speed of classical simulation could be a bottleneck for large qubit numbers, the time requirements remain manageable at the current scale of our benchmark. Additionally, quantum algorithm design is not inherently dependent on the number of qubits. For instance, the structure of the Deutsch-Jozsa algorithm - where Hadamard gates are applied before and after the oracle - remains consistent regardless of qubit count. In this sense, simulations and verifications on smaller qubit instances can effectively reveal the model’s capabilities for a given algorithm to a meaningful extent.
- **A few clarifications issues**: Detailed clarifications have been made in individual response to the reviewers.

[1] Z. Ye, Y. Huang, L. Li, and Y. Wang. Query complexity of generalized simon’s problem. Information and Computation, 281:104790, 2021.

[2] Z. Wu, D. Qiu, J. Tan, H. Li, and G. Cai. Quantum and classical query complexities for generalized simon’s problem. Theoretical Computer Science, 924:171–186, 2022.

---

### Author Response · Authors · 2024-11-21

Dear Reviewers,

We would like to express our gratitude for the time and effort you have dedicated to reviewing our paper. We have carefully considered your feedback and provided a detailed response in rebuttal. We are eager to hear your opinions and to address any further concerns you may have.

Thank you once again for your dedication, and we look forward to your response.

---

### Meta-Review · Area_Chair_6jN8 · 2024-12-22

**Metareview:**

The paper proposes a dataset to test AI capabilities to design and implementing different quantum algorithm in the form of quantum circuits. The authors also fine-tune a model on this dataset and analyze the performance.
It is unclear, what audience this approach is for. It does not show any principal limitations of existing LLM (it did not see the QC domain -> it is not performing well, so lets fine-tune). It is also not combined with other methodologies and is left with a smart prompting of existing models.

It also does not provide any new insights (faster/better/new) algorithms in the quantum computing field.
So, I would consider this work incremental, and this concerns is shared by the reviewers

**Additional Comments On Reviewer Discussion:**

The discussion was not very active. All the reviewers shared similar concerns, and although the authors have provided additional experiments, it was not enough to convince them to change their scores.

---

### Decision · Program_Chairs · 2025-01-22

Reject